https://doi.org/10.1038/s41467-021-22320-y · OPEN

# Protein kinase A controls the hexosamine pathway by tuning the feedback inhibition of GFAT-1

Sabine Ruegenberg [1,2], Felix A. M. C. Mayr [1], Ilian Atanassov[1], Ulrich Baumann[2] & Martin S. Denzel [1,3,4✉]

The hexosamine pathway (HP) is a key anabolic pathway whose product uridine 5'-diphospho-N-acetyl-D-glucosamine (UDP-GlcNAc) is an essential precursor for glycosylation processes in mammals. It modulates the ER stress response and HP activation extends lifespan in *Caenorhabditis elegans*. The highly conserved glutamine fructose-6-phosphate amidotransferase 1 (GFAT-1) is the rate-limiting HP enzyme. GFAT-1 activity is modulated by UDP-GlcNAc feedback inhibition and via phosphorylation by protein kinase A (PKA). Molecular consequences of GFAT-1 phosphorylation, however, remain poorly understood. Here, we identify the GFAT-1 R203H substitution that elevates UDP-GlcNAc levels in *C. elegans*. In human GFAT-1, the R203H substitution interferes with UDP-GlcNAc inhibition and with PKA-mediated Ser205 phosphorylation. Our data indicate that phosphorylation affects the interactions of the two GFAT-1 domains to control catalytic activity. Notably, Ser205 phosphorylation has two discernible effects: it lowers baseline GFAT-1 activity and abolishes UDP-GlcNAc feedback inhibition. PKA controls the HP by uncoupling the metabolic feedback loop of GFAT-1.

[1] Max Planck Institute for Biology of Ageing, Cologne, Germany. [2] Institute of Biochemistry, University of Cologne, Cologne, Germany. [3] CECAD - Cluster of Excellence, University of Cologne, Cologne, Germany. [4] Center for Molecular Medicine Cologne (CMMC), University of Cologne, Cologne, Germany. ✉email: martin.denzel@age.mpg.de

The hexosamine pathway (HP) converts fructose-6-phosphate (Frc6P) to uridine 5′-diphospho-*N*-acetyl-D-glucosamine (UDP-GlcNAc) (Fig. 1a)[1]. In all, 2 to 3 % of cellular glucose enter the HP, as well as L-glutamine (L-Gln), acetyl-coenzyme A, and uridine[2]. Hence, the HP integrates sugar, amino acid, fatty acid, and nucleotide metabolism and is considered an important nutrient sensing pathway. The GlcNAc moiety of UDP-GlcNAc is used as a building block for several macromolecules, including peptidoglycans in bacteria, chitin in fungi and insects, or glycosaminoglycans such as hyaluronic acid in vertebrates[3–8]. Moreover, UDP-GlcNAc is an essential precursor for glycosylation reactions in eukaryotes. N-linked

| Gene ID | Gene name | Position | Codon | Amino acid |
|---------|-----------|----------|-------|-----------|
| C07E3.2 | *pro-2* | II10340139 | cAt / cCt | H / P |
| ZK666.7 | *clec-61* | II10489388 | cGa / cAa | R / Q |
| W02B12.8 | *rga-1* | II11468055 | cTc / cGc | L / R |
| F07A11.2 | *gfat-1* | II11594718 | cGt / cAt | R / H |
| ZK930.5 | *ZK930.5* | II11882734 | aGa / aCa | R / T |

WT / *dh783*

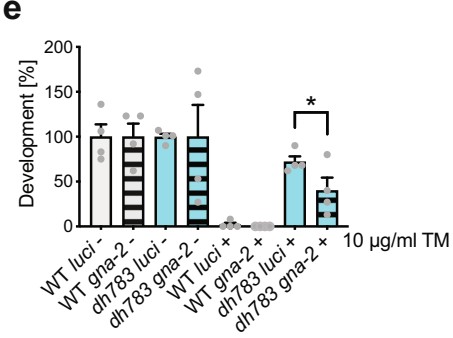

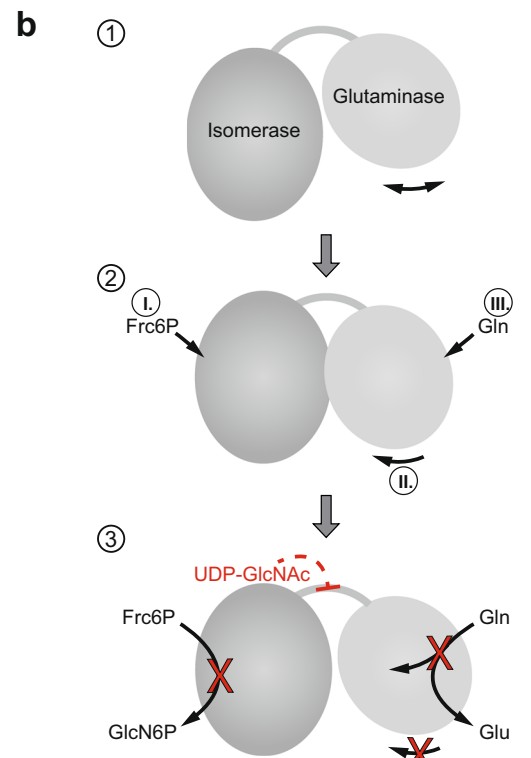

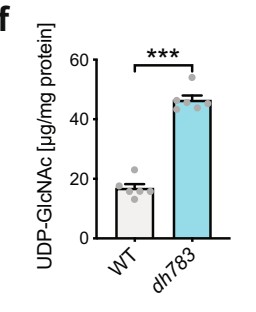

**Fig. 1 Characterization of *gfat-1(dh783) C. elegans* mutants. a** Schematic representation of the hexosamine pathway (green box). The enzymes in the pathway are glutamine fructose-6-phosphate amidotransferase (GFAT-1/-2), glucosamine-6-phosphate *N*-acetyltransferase (GNA-1), phosphoglucomutase (PGM-3), UDP-*N*-acetylglucosamine pyrophosphorylase (UAP-1), and glucosamine-6-phosphate deaminase (GNPDA-1/-2). UDP-GlcNAc inhibits eukaryotic GFAT (red line). **b** Catalytic scheme of one GFAT monomer. (1) Before catalysis: The glutaminase domain does not adopt a fixed position. (2) Substrate binding: I. Frc6P binds, II. the glutaminase domain adopts a specific position, III. L-Gln binds. (3) Catalysis and UDP-GlcNAc inhibition. Catalysis: L-Gln is hydrolyzed to L-Glu and the released ammonia is shuttled through an ammonia channel from the glutaminase to the isomerase domain. There, Frc6P is isomerized to Glc6P and the ammonia is transferred to build GlcN6P. UDP-GlcNAc inhibition: UDP-GlcNAc binds to the isomerase domain, interacts with the interdomain linker, and inhibits the glutaminase function and thereby the GlcN6P production. **c** *C. elegans* (N2 wild type (WT) and *gfat-1(dh783)*) developmental resistance assay on NGM plates containing 10 μg/ml tunicamycin (TM) (mean + SEM, $n = 5$, *** $p < 0.0001$, unpaired two-sided *t* test). **d** Frequency plot of normalized parental alleles on chromosome II of *dh783*. The CloudMap Hawaiian Variant Mapping with WGS tool displays regions of linked loci where pure parental allele SNP positions instead of allele positions containing Hawaiian SNPs are over-represented. Gray bars represent 1 Mb and red bars represent 0.5 Mb sized bins. Table: Candidate non-synonymous SNPs between 10 and 12 Mb on chromosome II of *gfat-1(dh783)* animals. **e** *C. elegans* (N2 wild type (WT) and *gfat-1(dh783)*) developmental resistance assay on NGM plates containing 10 μg/ml tunicamycin (TM) upon RNAi treatment targeting *gna-2* and control *luciferase* (*luci*) (mean + SEM, $n = 4$, * $p = 0.0437$, unpaired one-sided *t* test). **f** UDP-GlcNAc levels normalized to protein content in N2 wild type (WT) and *gfat-1(dh783)* animals (mean + SD, $n = 5$, *** $p < 0.0001$, unpaired two-sided *t* test). Source data are provided as a Source Data file.

glycosylation takes place in the endoplasmic reticulum (ER) and is crucial for proper protein folding and the solubility of proteins[9]. Mucin-type O-linked glycosylation occurs in the Golgi apparatus and is found on many cell surface and secreted proteins[10]. In protein O-GlcNAcylation that occurs in cytoplasm and nucleus, UDP-GlcNAc is the donor for the attachment of a single GlcNAc moiety to serine or threonine residues[11]. This process can fine-tune a protein's function akin to phosphorylation and often O-GlcNAcylation and phosphorylation compete for modification of the same sites[12,13].

HP activity is regulated by its first and rate-limiting enzyme glutamine fructose-6-phosphate amidotransferase (GFAT, EC 2.6.1.16)[2]. Two GFAT paralogs exist that primarily differ in their tissue-specific expression patterns[14]. GFAT is a modular enzyme composed of a glutaminase domain responsible for hydrolysis of L-Gln into L-glutamate (L-Glu) and an isomerase/transferase domain that catalyzes the isomerization of Frc6P to glucose-6-phosphate (Glc6P) as well as the transfer of ammonia to Glc6P to build glucosamine-6-phosphate (GlcN6P)[15]. The relative orientation of the two domains with their respective active sites is relevant for catalysis and affected by substrate binding. In the absence of Frc6P, the glutaminase domain is very flexible (Fig. 1b, 1) and no corresponding electron density is visible in the *E. coli* GFAT structure, although SDS-PAGE analyses of dissolved crystals indicate the presence of the full-length protein[16,17]. The reaction is initiated by binding of Frc6P to the isomerase domain, which triggers the flexible glutaminase domain to adopt a specific position relative to the isomerase domain and locks the two domains in this state[16] (Fig. 1b, 2). Subsequently, L-Gln binds to the gluta-minase domains, closing the pocket of the glutaminase active site[18]. This is accompanied by a rotation of the glutaminase domain by 21° relative to the isomerase domain[19] (Fig. 1b, 2). Finally, a solvent-inaccessible channel is formed that links the two active sites to allow the diffusion of ammonia[19–21].

A disturbed function of GFAT-1 due to *gfat-1* mutations cause limb-girdle congenital myasthenic syndrome with tubular aggregates[22,23]. This inherited disorder is characterized by defective neuromuscular transmission through impaired neuromuscular junctions, which transmit impulses from motor neurons to skeletal muscle fibers[22]. Moreover, many studies suggest a role of the HP and GFAT in the development and progression of diabetes or cancer[24–26]. Both disorders are characterized by an aberrant glucose metabolism, and the HP links altered metabolism with aberrant glycosylation. Especially elevated O-GlcNAcylation contributes to the pathogenesis of diabetes and cancer. Altered O-GlcNAcylation was also reported to play a critical role in neurodegenerative diseases, heart disease, and inflammation[27,28].

Given that GFAT drives the metabolic flux of the HP, its regulation is of great physiological importance and in eukaryotes it

occurs through UDP-GlcNAc feedback inhibition[29–31] and through phosphorylation[32,33]. In human GFAT-1, Ser205 and Ser235 are phosphorylated by cAMP-dependent protein kinase A (PKA)[32,34,35]. Ser205 phosphorylation has reported effects on GFAT-1 activity, although the published data are contradictory: On the one hand, phosphorylation of Ser205, and the corresponding Ser202 in human GFAT-2, were reported to increase activity[32,35]. On the other hand, it was described that Ser205 phosphorylation leads to decreased GFAT-1 activity[34]. Phosphorylation of the second PKA site Ser235 in GFAT-1, which is not conserved in all eukaryotes and is absent in GFAT-2, seems not to affect the enzymatic activity[34,36]. Ser243 was shown to be phosphorylated by adenosine monophosphate (AMP)-activated protein kinase (AMPK) and calcium/calmodulin-dependent kinase II, but the effect on GFAT activity is not clear: depending on the study, phosphorylation at this site is reported to be either activating or inhibiting[33,37,38].

Previously, in a *Caenorhabditis elegans* forward genetic muta-genesis screen we identified gain-of-function mutations in *gfat-1* that suppress tunicamycin-induced proteotoxic stress[39]. These mutations result in single amino acid substitutions in GFAT-1 and increase UDP-GlcNAc concentrations in the worms that also show improved protein quality control as well as a significant lifespan extension[39]. Recently, we found that loss of regulation by UDP-GlcNAc feedback inhibition is one gain-of-function mechanism and identified a critical role of the interdomain linker in UDP-GlcNAc inhibition[40]. Although UDP-GlcNAc binds to the isomerase domain of GFAT-1, this interaction inhibits the glutaminase function and consequently the combined GlcN6P production[41]. We proposed that UDP-GlcNAc disturbs the tight coupling of the active sites by interacting with the interdomain linker and by interfering with the relative orientation of the two domains[40] (Figs. 1b, 3). Additional analyses of other GFAT-1 gain-of-function mutations might help to understand the molecular mechanisms of GFAT-1.

Here, we identify the GFAT-1 R203H gain-of-function mutation that interferes with UDP-GlcNAc inhibition and with PKA-dependent phosphorylation at Ser205. Analyses of the phospho-mimic S205D substitution resolved the controversially discussed effect of Ser205 phosphorylation. Our data demonstrate that PKA phosphorylation at Ser205 lowers GFAT-1 activity and simultaneously blocks UDP-GlcNAc inhibition. We propose a model in which Ser205 phosphorylation controls domain interactions in GFAT-1 to modulate activity and feedback inhibition.

## Results

**The *gfat-1(dh783)* gain-of-function point mutant is resistant to ER stress.** Previously, we performed a forward genetic screen in

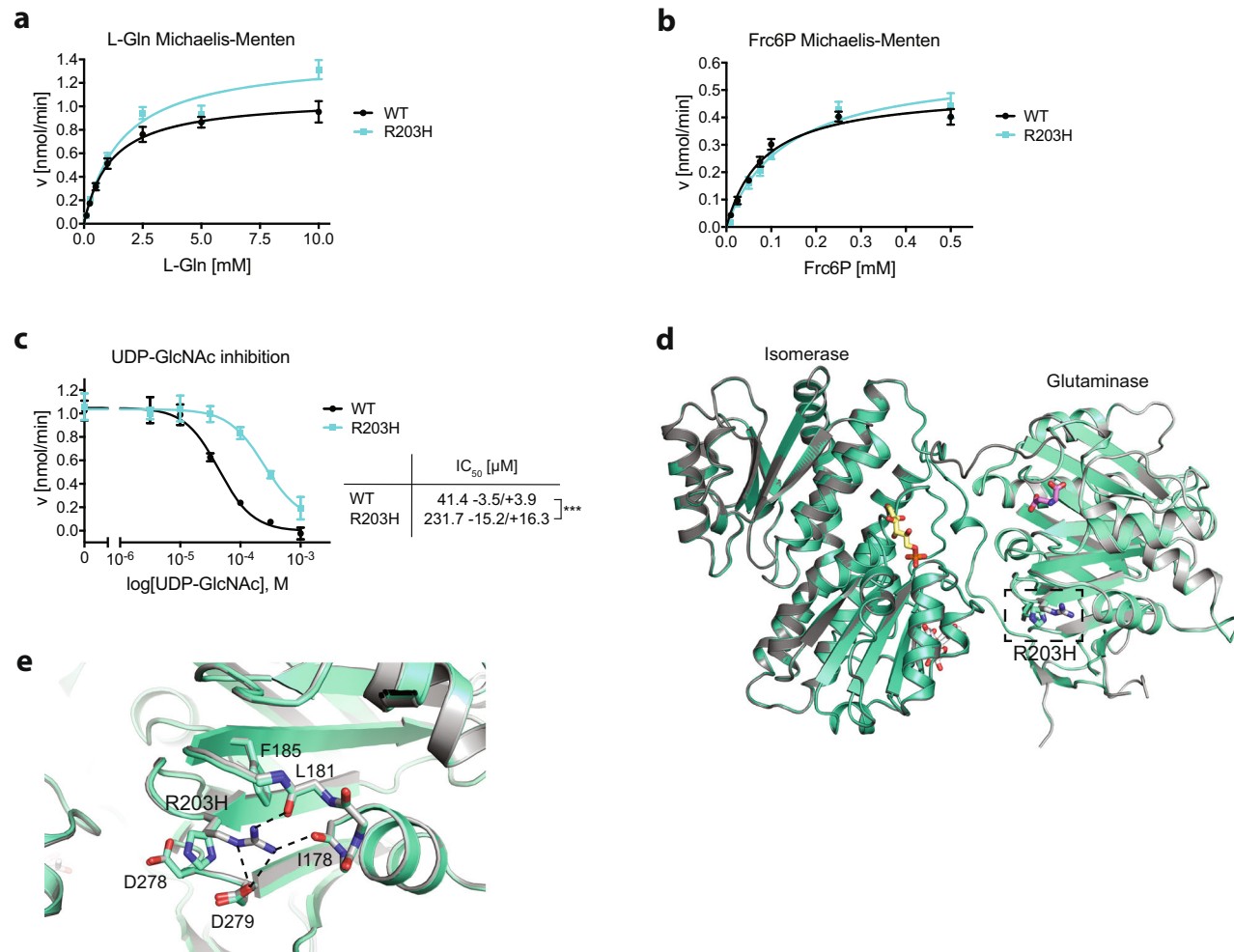

**Fig. 2 The GFAT-1 R203H gain-of-function substitution perturbs UDP-GlcNAc feedback inhibition. a** L-Gln kinetic of wild type (WT, black circle) and R203H (cyan square) GFAT-1 (mean ± SEM, WT $n = 5$, R203H $n = 4$). **b** Frc6P kinetic of wild type (WT, black circle) and R203H (cyan square) GFAT-1 (mean ± SEM, WT $n = 5$, R203H $n = 4$). **c** Representative UDP-GlcNAc dose response assay of wild type (WT, black circle) and R203H (cyan square) GFAT-1 (mean ± SD, $n = 3$). Table: IC$_{50}$ UDP-GlcNAc values (mean ± SEM, $n = 4$, ***$p < 0.0001$, unpaired two-sided $t$ test). **d**, **e** Position of the R203H mutation in the structure of GFAT-1. Proteins are presented as cartoons. Superposition of wild type GFAT-1 (light gray/dark gray, PDB ID 6R4E) and R203H GFAT-1 (green–cyan, teal) with an RMSD of 0.38 Å over 1280 main chain residues. Glc6P (yellow sticks), L-Glu (violet sticks), and UDP-GlcNAc (white sticks) are highlighted, as well as the position of R203H (black box). **d** Overview. R203H (sticks) is located at the glutaminase domain of GFAT-1 (dashed box). **e** Close-up view of the position of R203H focusing on residues in close proximity. The R203H mutation and residues in close proximity to the mutation are highlighted with sticks. Arg203 interacts with the neighboring loops (dashed lines). Source data are provided as a Source Data file.

*C. elegans* for mutants that are resistant to tunicamycin-induced proteotoxic stress[39]. From this screen, we obtained the mutant allele *dh783*, which was not characterized until now. While 10 μg/ml tunicamycin was toxic for N2 wild type worms, the *dh783* allele conferred strong tunicamycin resistance (Fig. 1c and Supplementary Fig. 1a). To identify the causal mutation, we performed Hawaiian single nucleotide polymorphism (SNP) mapping[42]. The normalized linkage score identified SNPs in exons of five candidate genes on chromosome II including *gfat-1* (Fig. 1d). To test an involvement of the HP in the resistance phenotype, we performed a developmental tunicamycin resistance assay and knocked down the expression of glucosamine-6-phosphate *N*-acetyltransferase (Fig. 1a), the enzyme catalyzing the second HP reaction. HP inhibition by RNAi targeting the *C. elegans* homolog *gna-2* reduced the developmental tunicamycin resistance observed in the *dh783* mutants (Fig. 1e). Given that tunicamycin toxicity can be suppressed by elevated UDP-GlcNAc levels through gain-of-function mutations in *gfat-1*[39], we quantified UDP-GlcNAc and its epimer UDP-GalNAc by mass spectrometry. Indeed, we found significantly elevated UDP-GlcNAc

and UDP-GalNAc levels in whole worm lysates in mutant carriers of the *dh783* allele (Fig. 1f and Supplementary Fig. 1b). We conclude that tunicamycin resistance was caused by the *gfat-1 (dh783)* allele due to the GFAT-1 R203H gain-of-function substitution.

**The GFAT-1 R203H substitution interferes with UDP-GlcNAc inhibition leading to gain-of-function.** GFAT-1 is well-conserved from *C. elegans* to humans (Supplementary Fig. 2a). To decipher the effect of the R203H substitution, we characterized human GFAT-1 R203H in activity assays and found similar kinetics for L-Glu and GlcN6P production as in wild type GFAT-1 (Fig. 2a, b and Table 1). In contrast, UDP-GlcNAc dose response assays revealed an ~6-fold higher IC$_{50}$ value of R203H GFAT-1 compared to wild type GFAT-1 (R203H: 231.7 −15.2/+16.3 μM; wild type: 41.4 −3.5/+3.9 μM) (Fig. 2c). Subsequently, we crystallized human GFAT-1 R203H to further elucidate the structural basis of the reduced UDP-GlcNAc sensitivity. The GFAT-1 R203H crystals diffracted to a resolution limit of 2.77 Å (Table 2).

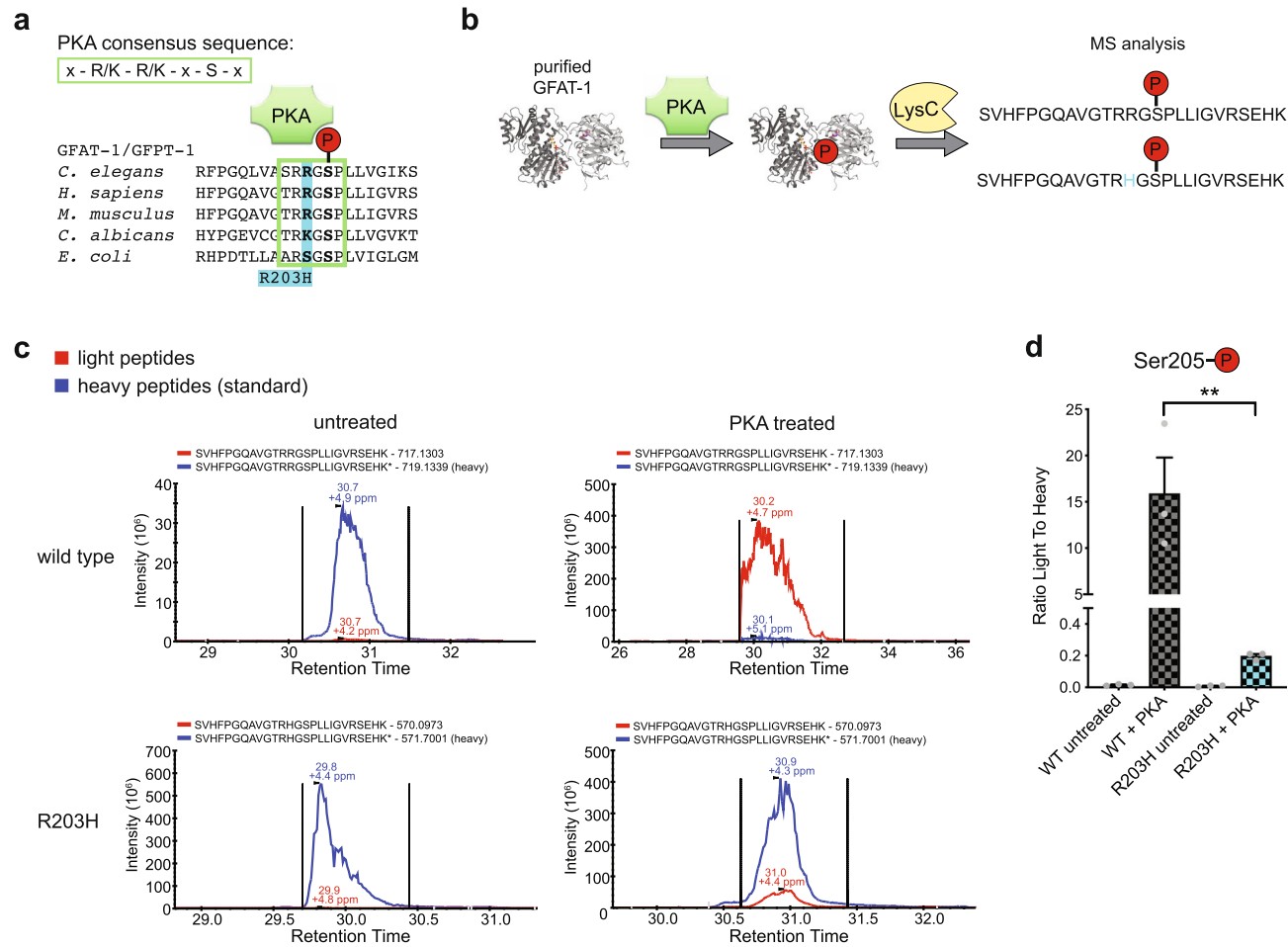

**Fig. 3 The GFAT-1 gain-of-function substitution R203H disturbs PKA phosphorylation at Ser205. a** Position of gain-of-function mutation R203H in a protein sequence alignment of GFAT-1. R203H (cyan) disrupts the PKA consensus sequence (green box) at Ser205. **b** Workflow for the in vitro phosphorylation analysis. **c** Representative quantification of the light and heavy peptides by mass spectrometry. **d** Quantification of the Ser205 phosphorylation of wild type (WT, gray) and R203H (cyan) GFAT-1 before and after treatment with PKA (mean + SEM, $n = 4$, ** $p = 0.0021$, one-way ANOVA). Source data are provided as a Source Data file.

**Table 1 Kinetic parameters.**

| | L-Glu production | | | D-GlcN6P production | | |
|---|---|---|---|---|---|---|
| | $K_m$ L-Gln [mM] | $k_{cat}$ [sec$^{-1}$] | $k_{cat}/K_m$ [mM$^{-1}$ sec$^{-1}$] | $K_m$ Frc6P [mM] | $k_{cat}$ [sec$^{-1}$] | $k_{cat}/K_m$ [mM$^{-1}$ sec$^{-1}$] |
| Wild type | 1.1 ± 0.2 | 3.6 ± 0.2 | 3.3 | 0.08 ± 0.01 | 1.7 ± 0.1 | 21.3 |
| R203H | 1.6 ± 0.27 | 4.8 ± 0.26 | 3.0 | 0.13 ± 0.02 | 2.0 ± 0.15 | 15.4 |
| S205D | 1.3 ± 0.23 | 1.5 ± 0.08 | 1.2 | 0.18 ± 0.04 | 0.8 ± 0.07 | 4.4 |
| L405R | 1.0 ± 0.10 | 3.0 ± 0.08 | 3.0 | 0.07 ± 0.02 | 1.9 ± 0.15 | 27.1 |

Compared to wild type GFAT-1, the R203H substitution does not cause major structural changes (Fig. 2d, e and Supplementary Fig. 2b). While Arg203 interacts with the backbone residues of Ile178 and Leu181, as well as with Asp279 in wild type GFAT-1, these interactions are abolished in the R203H GFAT-1 (Fig. 2e). In the mutant enzyme, however, the side chain of His203 protrudes between Asp278 and Asp279, without affecting the orientation of the neighboring loop (residues 277–280) (Fig. 2e). Together these findings suggest that blunted UDP-GlcNAc feedback inhibition is responsible for the gain-of-function of GFAT-1 R203H and that Arg203 has a functional role in UDP-GlcNAc inhibition.

**The GFAT-1 R203H substitution interferes with PKA phosphorylation at Ser205.** In addition to the observed gain-of-

function caused by reduced feedback inhibition, the R203H substitution alters the PKA consensus sequence of Ser205 from RRGS to RHGS, suggesting a potential change of Ser205 phosphorylation and a subsequent loss of regulation (Fig. 3a). To test whether GFAT-1 R203H can be phosphorylated at Ser205, we purified wild type and R203H GFAT-1 and performed in vitro phosphorylation with PKA, followed by LysC protease digest and untargeted as well as targeted mass spectrometric analyses (Fig. 3b). In the untargeted analysis, we identified a number of phosphorylated sites, among them the PKA phosphorylation sites Ser205 and Ser235[32,34] (Supplementary Dataset 1). For the targeted analysis, heavy isotope labeled phosphorylated peptides of wild type or R203H GFAT-1 were spiked in, allowing more sensitive relative quantification of the Ser205 phosphorylation

**Table 2 Data collection and refinement statistics.**

| | GFAT-1 R203H | GFAT-1 L405R | GFAT-1 S205D |
|---|---|---|---|
| Data collection | | | |
| Wavelength (Å) | 1.00 | 1.00 | 0.97 |
| Space group | P 4₁2₁2 | P 4₁2₁2 | P 4₁2₁2 |
| Unit cell dimensions | | | |
| $a$, $b$, $c$ (Å) | 152.8 | 152.9 | 153.6 |
| | 152.8 164.8 | 152.9 166.2 | 153.6 166.1 |
| α, β, γ (°) | 90 90 90 | 90 90 90 | 90 90 90 |
| Resolution range (Å) | 48.65 –2.77 | 48.74–2.38 | 73.07–2.22 |
| | (2.87– 2.77) | (2.47–2.38) | (2.30 – 2.22) |
| $R_{merge}$ (%) | 22.0 (230.5) | 11.1 (202.3) | 9.6 (255.2) |
| $I / \sigma I$ | 14.05 (1.16) | 20.15 (1.23) | 22.99 (1.43) |
| $CC_{1/2}$ (%) | 99.7 (45.0) | 99.9 (50.1) | 100 (58.1) |
| Completeness (%) | 99.8 (97.9) | 99.9 (99.2) | 99.5 (95.4) |
| Redundancy | 13.3 (12.8) | 13.4 (13.3) | 25.8 (26.5) |
| Refinement | | | |
| Reflections used in refinement | 49,787 | 78,970 | 97,286 |
| Reflections used for R-free | 1993 | 1983 | 1933 |
| $R_{work}$ / $R_{free}$ (%) | 20.1/23.9 | 19.3/21.1 | 17.7/19.4 |
| Number of non-hydrogen atoms | 10,329 | 10,529 | 10,562 |
| Macromolecules | 10,277 | 10,384 | 10,387 |
| Ligands | 32 | 32 | 32 |
| Solvent | 20 | 113 | 143 |
| Average B-factor (Å²) | 95.73 | 84.61 | 79.58 |
| Macromolecules | 95.97 | 85.11 | 79.96 |
| Ligands | 52.14 | 49.82 | 50.92 |
| Solvent | 42.46 | 48.53 | 58.74 |
| R.m.s. deviations | | | |
| Bond lengths (Å) | 0.002 | 0.002 | 0.003 |
| Bond angles (°) | 0.45 | 0.45 | 0.49 |
| Ramachandran favored (%) | 95.9 | 96.3 | 96.8 |
| Ramachandran allowed (%) | 4.0 | 3.5 | 3.2 |
| Ramachandran outliers (%) | 0.08 | 0.23 | 0.15 |
| Rotamer outliers (%) | 0.09 | 0.26 | 0.44 |
| Clashscore | 0.78 | 0.67 | 0.67 |
| PDB code | 6ZMJ | 6ZMK | 7NDL |

Statistics for the highest-resolution shell are shown in parentheses.

(Fig. 3b). Untreated insect cell-derived wild type and R203H GFAT-1 preparations showed low detectable levels of phosphorylation at Ser205. In vitro PKA treatment resulted in phosphorylation of wild type GFAT-1 at Ser205 (Fig. 3c, d). In contrast, the increase of Ser205 phosphorylation was 80-fold reduced in the presence of the R203H substitution (Fig. 3c, d). The second known PKA phosphorylation site of GFAT-1 at Ser235[34] was used as a positive control. For this site, we found a significant increase in the abundance of phosphorylated peptides corresponding to Ser235 in both wild type and R203H GFAT-1, proving full activity of PKA in our setup (Supplementary Fig. 3). Thus, disruption of the PKA consensus sequence by the R203H substitution prevents PKA-mediated phosphorylation at Ser205. Taken together, in addition to a lower sensitivity to UDP-GlcNAc inhibition, the GFAT-1 R203H substitution interferes with PKA-dependent regulation by suppressing Ser205 phosphorylation.

**PKA phosphorylation at Ser205 modulates UDP-GlcNAc inhibition of GFAT-1.** The loss of PKA phosphorylation in the GFAT-1 gain-of-function R203H variant suggests that Ser205 phosphorylation might be inhibitory. Until now, the effect of phosphorylation at Ser205 has been controversially reported as either activating[32] or inhibiting[34]. We aimed to mechanistically understand the effect of Ser205 phosphorylation and generated a GFAT-1 S205D mutant to mimic the phosphorylation. Importantly, wild type insect cell-derived GFAT-1 showed almost no phosphorylation at Ser205 (Fig. 3c, d), permitting the direct comparison with the S205D variant in activity assays. We observed reduced $k_{cat}$ values for L-Glu and GlcN6P production resulting in lower catalytic efficiency ($k_{cat}/K_m$) for both active sites (Fig. 4a, b and Table 1). Overall GlcN6P synthesis was decreased ~5-fold in the phospho-mimic mutant ($k_{cat}/K_m$: 4.4 mM$^{-1}$ sec$^{-1}$) compared to wild type GFAT-1 ($k_{cat}/K_m$: 21.3 mM$^{-1}$ sec$^{-1}$) (Table 1). Strikingly, the dose-dependent inhibition by UDP-GlcNAc was completely abolished in the S205D variant (Fig. 4c). These data suggest that PKA-dependent phosphorylation at Ser205 reduces GFAT-1 baseline activity in vitro. At the same time, the S205D variant remained constitutively active even in the presence of UDP-GlcNAc levels that strongly inhibit wild type GFAT-1. Thus, the presence of UDP-GlcNAc influences whether the phosphorylation is activating or inhibiting in vivo: at low UDP-GlcNAc concentrations up to ~30 μM, the phosphorylation is inhibiting due to the reduced catalytic function, while at high UDP-GlcNAc concentrations above ~30 μM, the phosphorylation is activating due to the loss of feedback inhibition.

Understanding the effect of Ser205 PKA phosphorylation on GFAT-1 activity, we next analyzed its relevance in vivo. For that purpose, we generated HEK293 cell lines that stably overexpress either wild type, R203H, or S205D GFAT-1 with an internal His₆-tag (Fig. 4d) and measured UDP-GlcNAc and UDP-GalNAc levels in the cell lysates (Fig. 4e and Supplementary Fig. 4). The cells were cultured at high glucose concentrations of 25 mM, to avoid any glucose limitation. Strikingly, wild type GFAT-1 overexpression did not elevate UDP-GlcNAc or UDP-GalNAc concentrations. Likewise, R203H GFAT-1 cells did not show increased UDP-GlcNAc or UDP-GalNAc levels (Fig. 4e and Supplementary Fig. 4). This suggests that UDP-GlcNAc feedback inhibition is effective in fully suppressing the activity of wild type and R203H GFAT-1 even during overexpression. In contrast, we found significantly elevated levels of UDP-GlcNAc and UDP-GalNAc when the phospho-mimic S205D variant of GFAT-1 was overexpressed (Fig. 4e and Supplementary Fig. 4). These data support an interference of the S205D substitution with the UDP-GlcNAc inhibition, which rendered GFAT-1 constitutively active. To further investigate the consequence of the phospho-mimic mutation in a physiological context, we introduced the homologous S202D substitution by CRISPR/Cas9 in C. elegans. While the homozygous mutation was lethal, heterozygous carriers developed normally and were used for tunicamycin resistance assays. GFAT-1 S202D expression resulted in significant tunicamycin resistance, indicating elevated UDP-GlcNAc levels (Fig. 4f). In summary, our data suggest that PKA controls GFAT-1 activity by interference with its UDP-GlcNAc feedback inhibition.

**PKA phosphorylation at Ser205 modulates GFAT-1 domain interactions.** To analyze the molecular consequence of the S205D substitution in human GFAT-1, we solved the crystal structure at 2.22 Å resolution (Table 2). Further, we performed thermal shift assays to assess stability and obtained tryptophan fluorescence spectra to analyze potential domain movements. In the S205D crystal structure, we did not observe major structural differences compared to wild type GFAT-1 (Fig. 5a). Notably, the PKA target residue Ser205 is located in a loop of the glutaminase domain

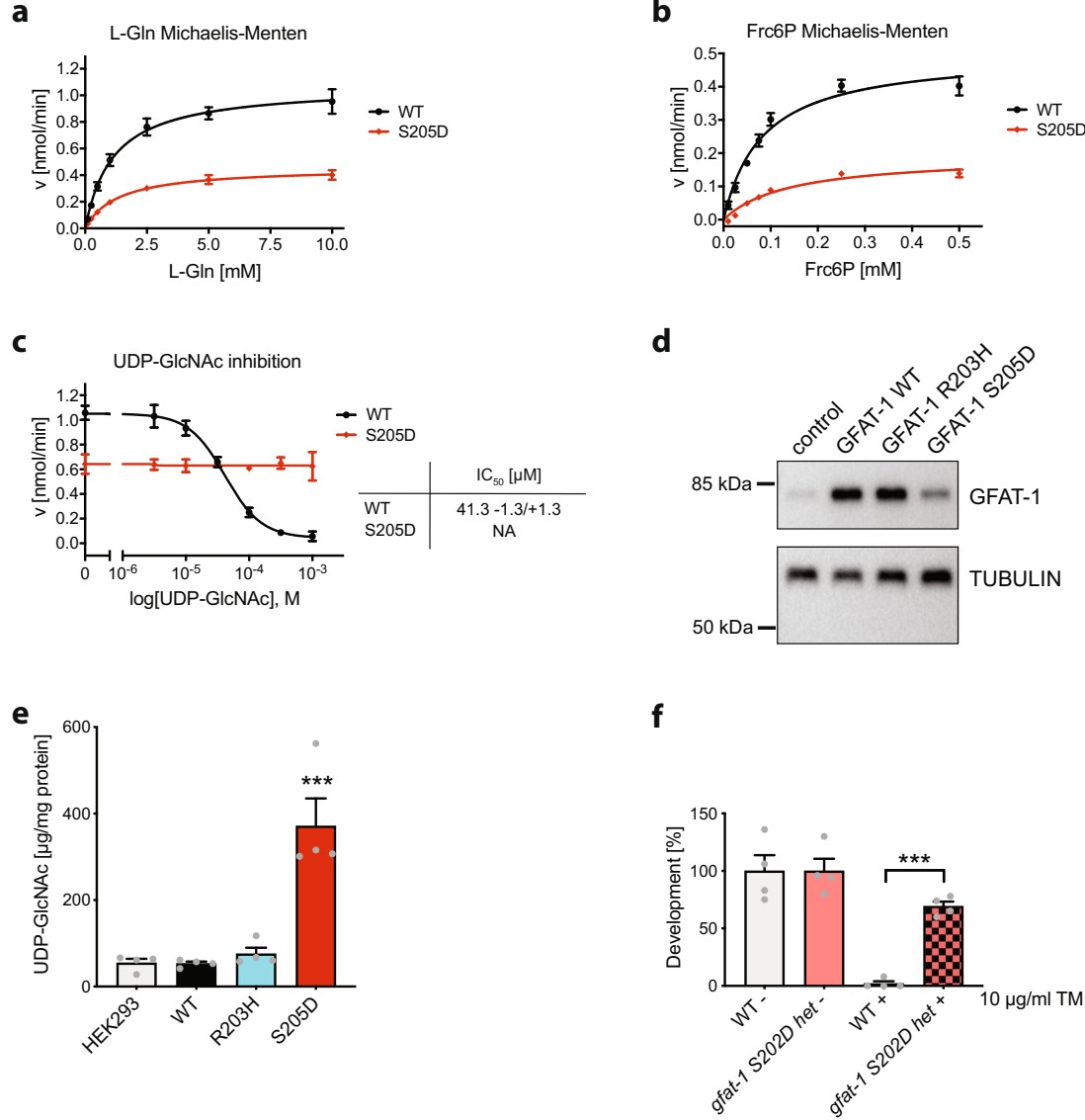

**Fig. 4 PKA phosphorylation at Ser205 modulates UDP-GlcNAc inhibition of GFAT-1. a** L-Gln kinetic of wild type (WT, black circles) and S205D (red diamonds) GFAT-1 (mean ± SEM, WT $n = 5$, S205D $n = 6$). **b** Frc6P kinetic of wild type (WT, black circles) and S205D (red diamonds) GFAT-1 (mean ± SEM, WT $n = 5$, S205D $n = 4$). **c** Representative UDP-GlcNAc inhibition of wild type (WT, black circles) and S205D (red diamonds) GFAT-1 (mean ± SD, $n = 3$). Table: $IC_{50}$ UDP-GlcNAc values (mean ± SEM, $n = 3$). **d** Representative Western blot analysis of GFAT-1 protein levels in control HEK293 cells and HEK293 cells stably overexpressing the indicated GFAT-1 variants. The experiment was repeated three times with similar results. **e** LC/MS measurement of UDP-GlcNAc normalized to protein content presented as means + SEM with $n = 4$, *** $p < 0.0001$ versus wild type (WT), one-way ANOVA. **f** *C. elegans* (N2 wild type (WT) and heterozygous (het) *gfat-1 S202D* (*syb3246*)) developmental resistance assay on NGM plates containing 10 µg/ml tunicamycin (TM) (mean + SEM, $n = 4$, *** $p < 0.0001$, unpaired two-sided *t* test). Source data are provided as a Source Data file.

pointing towards the glutaminase active site and the interdomain cleft between the glutaminase and isomerase domains (Fig. 5a). Ser205 is thus optimally positioned to influence domain interactions upon phosphorylation.

Next, we assessed the thermal stability of GFAT-1 in thermal shift assays. Full-length GFAT-1 showed two clearly distinguishable melting points at 53 and 65 °C, which are termed "low" and "high" hereafter (Fig. 5b and Supplementary Fig. 5a). Most likely, these two melting points represent the two domains, which possess different stabilities: the isolated isomerase domain melted at 64 °C suggesting that the high melting point corresponded to the isomerase domain, while the low melting point was assigned to the glutaminase domain (Fig. 5b and Supplementary Fig. 5a). To investigate their relative interactions, rising salt concentrations

were then used to destabilize the domains. For full-length GFAT-1, high salt (0.5–1.0 M) lowered the melting point of the glutaminase domain by 3–4 °C, and it lowered the melting point of the isomerase domain by 1.7–2 °C (Fig. 5c). In contrast, the isolated isomerase domain showed a much stronger reduction of the melting temperature of up to −10 °C (Fig. 5c). Together, these results show that in full-length GFAT-1 the isomerase domain is stabilized by the glutaminase domain, indicating domain interactions. We conclude that analysis of the melting points in GFAT-1 variants can reveal changes in the interaction between its domains. To independently test whether thermal shift assays can indeed indicate domain interactions, we generated the C-tail lock mutant L405R of human GFAT-1 and characterized it structurally, as well as in thermal shift assays and in kinetic

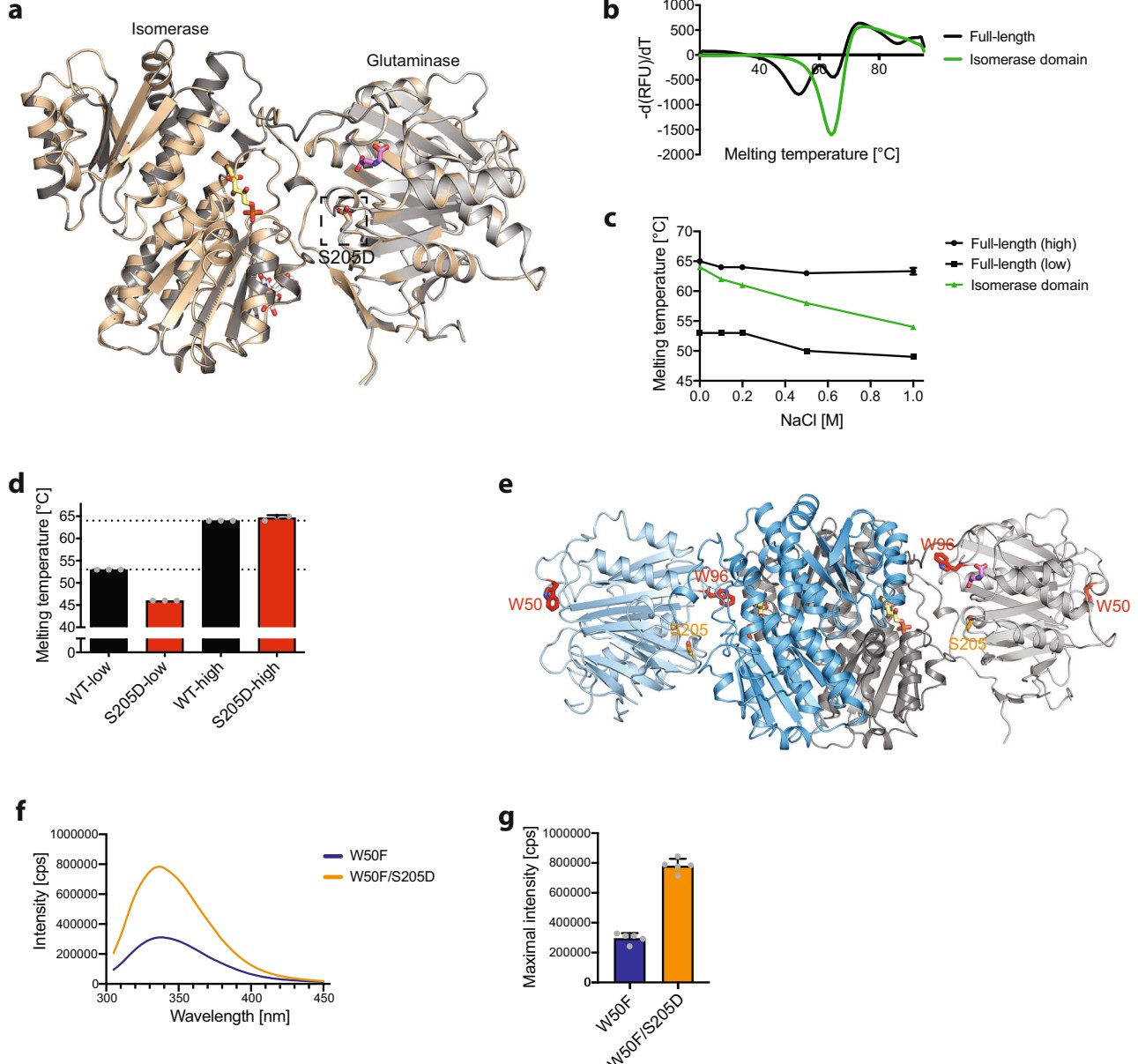

**Fig. 5 Altered domain interactions of GFAT-1 after PKA phosphorylation at Ser205. a** Position of the phosphorylation site Ser205 in the structure of GFAT-1. Proteins are presented as cartoons. Superposition of wild type GFAT-1 (gray, PDB ID 6R4E) and S205D GFAT-1 (wheat) with an RMSD of 0.77 Å over 1312 main chain residues. Glc6P (yellow sticks), L-Glu (violet sticks), and UDP-GlcNAc (white sticks) are highlighted, as well as the position of S205D (sticks, dashed box). **b** Representative derivative melting curves of full-length GFAT-1 (black) and the isolated GFAT-1 isomerase domain (green) in standard SEC buffer without NaCl. **c** Melting temperatures of full-length GFAT-1 (black) and the isolated GFAT-1 isomerase domain (green) GFAT-1 in SEC buffer with rising NaCl concentrations (mean + SD, $n = 3$). **d** Melting temperatures of wild type GFAT-1 (black) and GFAT-1 S205D (red) (mean + SD, $n = 3$). **e** Positions of tryptophan residues and Ser205 in the structure of wild type GFAT-1. The protein is presented as a cartoon (light gray/dark gray & light blue/marine, PDB ID 6R4E). Glc6P (yellow sticks) and L-Glu (violet sticks) are highlighted, as well as the positions of Trp50, Trp96, and Ser205 (sticks). **f** Representative fluorescence emission spectra of W50F (blue) and W50F/S205D (orange) GFAT-1. **g** Maximal fluorescence emission of W50F (blue) and W50F/S205D (orange) GFAT-1. (mean + SD, $n = 5$). Source data are provided as a Source Data file.

measurements. The well-conserved C-tail (C-terminal residues 670–681) is the major mobile element between the isomerase and glutaminase domain of GFAT. It bears residues of both active sites and mediates interactions between glutaminase and isomerase domain[19]. In *E. coli*, its flexibility is restricted by a salt-bridge between Arg331 and Glu608[43]. Miszkiel and Wojcie-chowski showed in molecular dynamic simulations that the C-tail lock mutation, which introduces a salt-bridge similar as observed in *E. coli*, limits the C-tail's flexibility[43]. We solved the crystal

structure (resolution 2.38 Å, Table 2) and confirmed that the L405R substitution indeed formed a salt-bridge with the C-terminal Glu681 and additionally interacted with Asp428 (Supplementary Fig. 5b–d). The kinetics (L-Gln and Frc6P) and the UDP-GlcNAc inhibition remained unaffected by the C-tail lock mutation (Table 1 and Supplementary Fig. 5e–g). Importantly, the L405R mutant clearly showed a single melting point at 56.9 ± 0.1 °C in thermal shift assays (Supplementary Fig. 5h, i), indicating a changed stability of both domains, which can only be

explained by the altered interactions between them. Therefore, thermal shift assays are a valid tool to analyze domain interactions of GFAT-1.

We used the thermal shift assay to test the role of the phospho-mimic S205D substitution in domain–domain interactions of GFAT-1. At physiological pH, phosphorylation introduces a di-anionic phosphate group that causes a repulsion with other negative charges and enables the formation of electrostatic interactions with positively charged residues. Compared to wild type GFAT-1, the S205D mutant showed strongly reduced thermal stability of the glutaminase domain, as indicated by the specific reduction of the lower melting point (Fig. 5d and Supplementary Fig 5j, k). The stability of the isomerase domain (high melting point) was not affected (Fig. 5d and Supplementary Fig 5j, k). Based on the crystal structure and circular dichroism (CD) measurements of wild type and S205D GFAT-1 (Fig. 5a and Supplementary Fig. 5l), we exclude major structural changes between wild type and S205D GFAT-1. Moreover, the crystallographic B-factors within the glutaminase domain of the phospho-mimic and the wild type structures were comparable (Supplementary Fig. 5m, n). Therefore, we rule out a destabilization of the glutaminase domain as a direct consequence of the S205D substitution. Instead, its reduced thermal stability is likely caused by altered interactions between the glutaminase and isomerase domains.

We further obtained fluorescence emission spectra to test if changed domain interactions or even a change in relative domain positions might occur in solution. While changes of the relative domain positions or stabilizing interactions were not detected in the S205D phospho-mimic crystal structure, they are conceivable in solution. Human GFAT-1 possesses two tryptophan residues: Trp50 is located at the surface of the glutaminase domain and Trp96 is positioned between the two domains at the glutaminase active site (Fig. 5e). Thus, the Trp96 fluorescence signal monitors the local environment within the interdomain cleft, potentially indicating changes in domain interactions. We mutated Trp50 to phenylalanine to eliminate its contribution to the fluorescence signal and measured the fluorescence spectra of wild type (W50F) or S205D (W50F/S205D) GFAT-1. The fluorescence signal of wild type (W50F) GFAT-1 was reduced compared to W50F/ S205D GFAT-1, pointing to a more solvent-exposed Trp96 in wild type GFAT-1 compared to the S205D variant (Fig. 5f, g). These data support the notion that the Ser205 phosphorylation alters the interaction of the two GFAT-1 domains, affecting activity and UDP-GlcNAc feedback inhibition.

## Discussion

Here we present the GFAT-1 R203H gain-of-function variant and decipher the two consequences of the substitution: first, GFAT-1 R203H displays strongly reduced PKA-dependent phosphorylation at Ser205 due to a disrupted consensus sequence and, second, shows weaker sensitivity to UDP-GlcNAc feedback inhibition. The current study resolves the contradicting reports regarding the effect of PKA phosphorylation at Ser205 in GFAT-1 and provides a model for phosphorylation-induced changes in domain interactions that control activity and regulation: the phospho-mimic S205D substitution lowered GFAT-1 activity while simultaneously abolishing UDP-GlcNAc inhibition. Importantly, we demonstrate a modulation of the UDP-GlcNAc feedback inhibition of GFAT-1 by PKA.

We describe two independent gain-of-function mechanisms of GFAT-1 R203H. First, when UDP-GlcNAc levels are low, the reduced Ser205 phosphorylation in GFAT-1 R203H results in higher GFAT-1 activity compared to wild type GFAT-1 that is inhibited by Ser205 phosphorylation. Second, when UDP-GlcNAc levels are high, wild type GFAT-1 is under stronger feedback inhibition than GFAT-1 R203H, resulting in elevated

cellular UDP-GlcNAc levels in the mutant. The observation of high UDP-GlcNAc levels in whole worm lysates in the *gfat-1* (*dh783*) R203H mutant points to the predominant role of the defective feedback inhibition as the relevant gain-of-function mechanism.

Reduced UDP-GlcNAc sensitivity of GFAT-1 R203H indicates a functional role of Arg203 in the UDP-GlcNAc feedback mechanism. In wild type GFAT-1, Arg203 is implicated in a salt-bridge network, which might help to position the glutaminase and isomerase domains (Fig. 6a). Previously, we proposed that UDP-GlcNAc promotes a catalytically unfavorable domain orientation, inhibiting GFAT-1[40]. In general, domain movements are common in the family of glutamine amidotransferases and the function of GFAT depends on the relative orientation of the glutaminase and isomerase domains[44]. Mouilleron et al. reported a structure of the *E. coli* GFAT active site mutant C1A (PDB 3OOJ) with a drastically changed orientation of the glutaminase domain relative to the isomerase domain, where one monomer adopts an inactive orientation without forming the ammonia channel[45]. This structure underpins the high flexibility of the two domains relative to each other. In wild type *E. coli* GFAT, the glutaminase domain adopts a specific position relative to the isomerase domain after Frc6P binding[16]. The presence of Frc6P activates the glutaminase function of GFAT more than 100-fold in *E. coli* and 70-fold in *C. albicans* and it is very likely that this substrate-induced activation occurs in the human GFAT as well[18,41]. Also, L-Gln binding induces a rotation of the glutaminase domain relative to the iso-merase domain in *E. coli* GFAT[19]. These domain movements upon substrate binding are required to enable specific interactions between the isomerase and glutaminase domains that are necessary for catalysis. It is well-established that phosphorylation can introduce conformational changes within loops or domains through altered electrostatic interactions[46]. A well-studied example for a phosphorylation-induced change in domain orientation is rabbit muscle glycogen phosphorylase, whose domains are shifted by ~50° relative to each other after phosphorylation[47,48]. The phosphorylation of GFAT-1 at Ser205 would introduce, at physiological pH, a negatively charged phosphate group, potentially affecting ionic interactions in close proximity. Arg202 of the glutaminase domain and Glu425 of the isomerase domain likely form a salt-bridge between the two domains and this interaction might be affected upon Ser205 phosphorylation (Fig. 6a).

Although we did not observe major structural differences between wild type and S205D GFAT-1 in the crystal structures, thermal shift assays and fluorescence measurements, which are performed in solution, clearly distinguished the two variants. Thermal shift assays showed a destabilization of the glutaminase domain in the S205D mutant, indicating altered domain inter-actions. Further, the fluorescence spectra are consistent with changes in relative domain position upon phosphorylation. We propose that Ser205 phosphorylation suppresses UDP-GlcNAc inhibition by stabilizing the glutaminase and isomerase domains in a catalytically productive orientation (Fig. 6b). While this conformation does not permit the high catalytic rate of wild type GFAT-1, the fixed conformation of the domains would prevent a UDP-GlcNAc induced domain orientation that might be needed for inhibition.

Our data reveal a UDP-GlcNAc concentration-dependent effect on the activity of phospho-Ser205 GFAT-1. This explains the previous contradicting reports regarding GFAT-1 Ser205 phosphorylation: Zhou et al. observed an activation[32], while Chang et al. reported an inhibition[34]. In both studies, cells were treated with forskolin to activate PKA and the GFAT-1 activity was analyzed in cell lysates. These lysates contained unknown UDP-GlcNAc concentrations that are likely to have affected the results. Moreover, Chang et al. confirmed the inhibitory effect

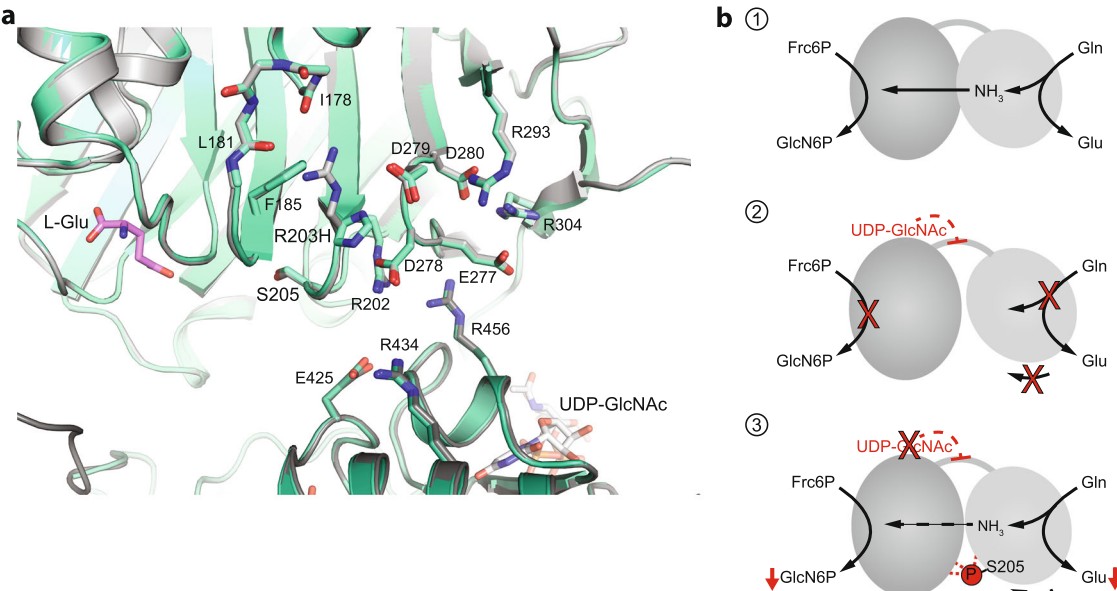

**Fig. 6 Altered domain interactions affect GFAT-1 activity. a** Superposition of the structures of wild type (gray) and R203H (green–cyan) GFAT-1 in cartoon representation. L-Glu (violet sticks) and UDP-GlcNAc (white sticks) are highlighted. Ser205, the R203H substitution, and residues that might form salt bridges in close proximity to the mutation are highlighted with sticks. **b** Schematic models of one GFAT monomer. (1) Catalysis: L-Gln is hydrolyzed to L-Glu and the released ammonia is shuttled through an ammonia channel from the glutaminase to the isomerase domain. (2) UDP-GlcNAc inhibition: UDP-GlcNAc binds to the isomerase domain, interacts with the interdomain linker, and inhibits the glutaminase function and thereby the GlcN6P production. (3) Ser205 phosphorylation: Upon phosphorylation, the catalytic activity is reduced and the UDP-GlcNAc inhibition is abolished.

in vitro with GST-tagged GFAT-1 in the absence of UDP-GlcNAc. However, the data obtained from a GST-tagged GFAT-1 should be taken with caution, because tagging GFAT at the N- or C-terminus interferes with catalysis[34,49,50].

Very few enzymes are known in which phosphorylation interferes with feedback inhibition[51]. The uncoupling of the feedback loop by GFAT-1 phosphorylation is an elegant mechanism to rapidly modulate its enzymatic activity after a stress-induced signal. The cAMP-PKA signaling pathway acts downstream of G protein-coupled receptors (GPCR) to mediate signals of neurotransmitters and hormones, such as glucagon or adrenaline[52–55]. The downstream protein targets of the cAMP-PKA signaling pathway regulate glucose homeostasis by inhibition of glycolysis and glycogen synthesis, as well as by triggering glucose release through stimulation of glycogenolysis and gluconeogenesis[55]. Our previous data indicated that GFAT-1 is under a constant UDP-GlcNAc inhibition in vivo[40]. Presumably, uncoupling of the feedback inhibition maintains GFAT-1 activity and ensures steady UDP-GlcNAc production when PKA activity is high. Given that UDP-GlcNAc is an essential building block and precursor for all glycosylation reactions in mammals, this mechanism is optimally positioned to ensure a constant UDP-GlcNAc supply. In all, our findings illuminate how the different means of GFAT-1 regulation, kinase signaling and feedback inhibition, are coordinated at the molecular level to fine-tune metabolic flux in the HP.

## Methods

***C. elegans* strains and culture**. *C. elegans* were maintained at 20 °C on nematode growth medium (NGM) agar plates with *E. coli* OP50[56]. The mutant was out-crossed to the wild type N2 strain. Strains used in this study: N2 (Bristol), AA2559 [*gfat-1(dh783)*], MSD544 [*gfat-1(syb3246)*] with the S202D substitution. The homozygous *syb3246* allele is lethal and mutants were maintained as a balanced heterozygous strain (*gfat-1(syb3246)/mIn1[mIs14 dpy-10(e128)*).

***C. elegans* developmental tunicamycin resistance assay**. Eggs were transferred to NGM plates containing 10 μg/ml tunicamycin (Sigma-Aldrich) seeded with

freeze-killed *E. coli*. Freeze-killed bacteria were obtained by three freeze/thaw cycles of pelleted overnight cultures. After 4 days, day 1 adults were scored, sick larvae were not counted. Throughout the experiment, strain identity was unknown to researchers.

**Mutant Hawaiian SNP mapping and sequence analysis**. Genomic DNA was prepared using the QIAGEN Gentra Puregene Kit according to the manufacturer's protocol. Whole genome sequencing was conducted on the Illumina HiSeq2000 platform. Paired-end 100 bp reads were used; the average coverage was larger than 16-fold. Sequences were analyzed using the CloudMap Hawaiian and Variant Discovery Mapping on Hawaiian Mapped Samples (and Variant Calling) Workflow_2-7-2014 pipeline on Galaxy[42,57]. The WS220/ce10 *C. elegans* assembly was used as reference genome.

**RNAi experiments**. HT115 *E. coli* carrying vectors for dsRNA of the target gene under an isopropyl β-D-1-thiogalactopyranoside (IPTG) inducible promotor were used. Bacteria were seeded on NGM plates containing 100 μg/μl ampicillin (Merck Millipore) and 1 mM IPTG (Roth), RNAi was done for two generations. RNAi targeting *luciferase* was used as control. All RNAi clones were obtained from Ahringer and Vidal RNAi libraries[58,59]. Clones were validated by Sanger sequencing.

**Small molecule LC/MS/MS analysis**. UDP-HexNAc concentrations were measured as described previously[39]. *C. elegans* or HEK293 cells were lysed in water by freeze/thaw cycles and subjected to chloroform/methanol extraction. Absolute UDP-HexNAc levels were determined using an Acquity UPLC connected to a Xevo TQ Mass Spectrometer (both Waters) and normalized to total protein content.

**Site-directed mutagenesis**. A pFL vector for the generation of baculoviruses for the expression of human GFAT-1 isoform 2 (hGFAT1) with internal His₆-tag between Gly299 and Asp300 was cloned previously[40]. The mutations W50F, R203H, S205D, and L405R were introduced into pFL-hGFAT1-His299 by site-directed mutagenesis as described previously[60]. Mutagenesis primers are listed in Supplementary Table 1.

**Baculovirus generation and insect cell expression of full-length GFAT-1**. Sf21 (DSMZ no. ACC 119) suspension cultures were maintained in SFM4Insect™ HyClone™ medium with glutamine (GE Lifesciences) in shaker flasks at 27 °C and 90 rpm. GFAT-1 variants were expressed using the MultiBac baculovirus system[61]. GFAT-1 variants were integrated into the baculovirus genome via Tn7 transposition and maintained as bacterial artificial chromosome in DH10EMBacY *E. coli*.

Recombinant baculoviruses were generated by transfection of *Sf21* cells with bacmid DNA. Obtained baculoviruses were used to induce protein expression in *Sf21* cells.

**Bacterial expression of GFAT-1 isomerase domain**. The isomerase domain of human GFAT-1 isoform 2 (residues 316–681) was integrated in the plasmid pET28a(+) using NdeI and HindIII restriction sites. Cloning primers are listed in Supplementary Table 1. This vector was used to recombinantly express the isomerase domain with N-terminal His$_6$tag and a thrombin cleavage site under the control of the T7 promoter in Rosetta (DE3) *E. coli*. LB cultures were incubated at 37 °C until an OD$_{600}$ of 0.4–0.6 was reached. Protein expression was induced by addition of 0.5 mM IPTG for 3 h at 37 °C. After harvest, pellets were stored at −80 °C.

**GFAT-1 purification**. *Sf21* cells (full-length GFAT-1) or *E. coli* (isomerase domain) were lysed by sonication in lysis buffer (50 mM Tris/HCl pH 7.5, 200 mM NaCl, 10 mM Imidazole, 2 mM Tris(2-carboxyethyl)phosphin (TCEP), 0.5 mM Na$_2$Frc6P, 10% (v/v) glycerol, supplemented with complete EDTA-free protease inhibitor cocktail (Roche), and 10 μg/ml DNAseI (Sigma)). Samples were cleared by centrifugation and the supernatant was loaded on Ni-NTA Superflow affinity resin (Qiagen). The resin was washed with lysis buffer and protein was eluted with lysis buffer containing 200 mM imidazole. For the isomerase domain, the His$_6$-tag was proteolytically removed using 5 U of thrombin (Sigma-Aldrich) per mg protein overnight at 4 °C, followed by an additional IMAC purification. Full-length GFAT-1 and the isomerase domain were further purified according to their size on a HiLoad™ 16/60 Superdex™ 200 prep grade prepacked column (GE Healthcare) using an ÄKTAprime chromatography system at 4 °C with a SEC buffer containing 50 mM Tris/HCl, pH 7.5, 2 mM TCEP, 0.5 mM Na$_2$Frc6P, and 10% (v/v) glycerol.

**Crystallization**. For crystallization experiments, the SEC buffer was supplemented with 50 mM L-Arg and 50 mM L-Glu to improve protein solubility[62]. GFAT-1 was crystallized at a concentration of 8 mg/ml in sitting-drops by vapor diffusion at 20 °C. The R203H and L405R GFAT-1 crystals grew in the PACT *premier*™ HT-96 (Molecular Dimensions) screen with a reservoir solution containing 0.1 M Bis-tris propane pH 8.5, 0.2 M potassium sodium tartrate tetrahydrate, and 20% (w/v) PEG3350 and the S205D GFAT-1 crystals grew in the PACT *premier*™ HT-96 (Molecular Dimensions) screen with a reservoir solution containing 0.1 M Bis-tris propane pH 8.5, 0.2 M sodium malonate dibasic monohydrate and 20% (w/v) PEG3350. Subsequent optimization screens were done with drop ratios of 1.5 μl protein solution to 1.5 μl precipitant solution and 2 μl protein solution to 1 μl precipitant solution. Data were collected from crystals cryoprotected with reservoir solution supplemented with 15% (v/v) glycerol.

**Data collection and refinement**. X-ray diffraction data were collected at beamline X06DA, Swiss Light Source, Paul Scherrer Institute, Villigen (Switzerland) with the DA+ software and at beamline P13, PETRA III, DESY, Hamburg (Germany) with the Hamburg version of mxCuBE. Data reduction and scaling was performed with XDS[63]. The mutant GFAT-1 structures were determined by molecular replacement with phenix.phaser[64,65] using the model of the wild type human GFAT-1 (PDB 6R4E, [https://doi.org/10.2210/pdb6R4E/pdb]) as search model. GFAT-1 was further manually built using COOT[66] and iterative refinement rounds were performed using phenix.refine[65]. Geometry restraints for ligands were generated with phenix.elbow[65]. Structures were visualized using PyMOL (Schrödinger).

**GDH-coupled activity assay and UDP-GlcNAc inhibition**. GFAT's amidohydrolyzing activity was measured in a coupled enzymatic assay using bovine glutamate dehydrogenase (GDH, Sigma-Aldrich G2626) as previously described[50] with small modifications. The reaction mixtures contained 6 mM Frc6P, 1 mM APAD, 1 mM EDTA, 50 mM KCl, 100 mM potassium phosphate buffer pH 7.5, 6.5 U GDH per well and (for L-Gln kinetics) varying concentrations of L-Gln. For UDP-GlcNAc inhibition assays the L-Gln concentration was 10 mM. The plate was pre-warmed at 37 °C and after enzyme addition, activity was monitored continuously at 363 nm in a SynergyHT (BioTek) microplate reader using Gen5 software (BioTek). Synthetized APADH was calculated with ε$_{(363\ nm,\ APADH)}$ = 9100 l mol$^{-1}$ cm$^{-1}$. Reaction rates were determined by Excel (Microsoft) and $K_m$, $v_{max}$, and IC$_{50}$ were obtained from Michaelis Menten or dose response curves, which were fitted by Prism 7 or 8 software (Graphpad).

**GNA-1 expression and purification**. The expression plasmid for human GNA-1 with N-terminal His$_6$-tag was cloned previously[40]. Human GNA-1 with N-terminal His$_6$-tag was expressed in Rosetta (DE3) *E. coli* cells. LB cultures were incubated at 37 °C until an OD$_{600}$ of 0.4–0.6 was reached. Protein expression was induced with 0.5 mM IPTG for 3 h at 37 °C. Harvested pellets were stored at −80 °C. GNA-1 purification protocol was adopted from Hurtado-Guerrero et al.[67] with small modifications. *E. coli* were lysed in 50 mM HEPES/NaOH pH 7.2, 500 mM NaCl, 10 mM imidazole, 2 mM 2-mercaptoethanol, 5% (v/v) glycerol with complete EDTA-free protease inhibitor cocktail (Roche), and 10 μg/ml DNAseI (Sigma) by sonication. The lysate was clarified by centrifugation and the supernatant loaded on Ni-NTA Superflow affinity resin (Qiagen). The resin was washed with wash buffer

(50 mM HEPES/NaOH pH 7.2, 500 mM NaCl, 50 mM imidazole, 5% (v/v) glycerol) and the protein was eluted with wash buffer containing 250 mM imidazole. Eluted protein was then dialyzed against storage buffer (20 mM HEPES/NaOH pH 7.2, 500 mM NaCl, 5% (v/v) glycerol).

**GNA-1 and GNA-1-coupled activity assays**. Human GNA-1 activity was measured as described previously[68]. For kinetic measurements, the assay mixture contained 0.5 mM AcCoA, 0.5 mM DTNB, 1 mM EDTA, 50 mM Tris/HCl pH 7.5, and varying concentrations of D-GlcN6P. The plates were pre-warmed at 37 °C and reactions were initiated by addition of GNA-1. Absorbance at 412 nm was followed continuously at 37 °C in a SynergyHT (BioTek) microplate reader. The amount of produced TNB, which matches CoA production, was calculated with ε$_{(412\ nm,\ TNB)}$ = 13,800 l mol$^{-1}$ cm$^{-1}$. Typically, GNA-1 preparations showed a $K_m$ of 0.2 ± 0.1 mM and a $k_{cat}$ of 41 ± 8 sec$^{-1}$.

GFAT's D-GlcN6P production was measured in a GNA-1-coupled activity assay following the consumption of AcCoA at 230 nm in UV transparent 96 well microplates[68]. The assay mixture contained 10 mM L-Gln, 0.1 mM AcCoA, 50 mM Tris/HCl pH 7.5, 2 μg hGNA-1, and varying concentrations of Frc6P. The plates were incubated at 37 °C for 4 min before adding L-Gln. Activity was monitored continuously at 230 nm and 37 °C in a SynergyHT (BioTek) microplate reader. The amount of consumed AcCoA was calculated with ε$_{(230\ nm,\ AcCoA)}$ = 6436 l mol$^{-1}$ cm$^{-1}$. As UDP-GlcNAc absorbs light at 230 nm, the GNA-1-coupled assay cannot be used to analyze UDP-GlcNAc effects on activity.

**Alignments**. The protein sequence alignment was created with following UnitProt IDs: *Caenorhabditis elegans*: Q95QM8, *Homo sapiens* isoform 2: Q06210-2 and formatted with the ESPript3 server (esprit.ibcp.fr/)[69]. For the structural superposition, the wild type human GFAT-1 structure in complex with L-Glu and Glc6P (PDB 6R4E, [https://doi.org/10.2210/pdb6R4E/pdb]) was aligned with the structures of mutant GFAT-1 R203H, S205D, and L405R. UDP-GlcNAc was displayed after superposition with the structure of wild type human GFAT-1 in complex with L-Glu, Glc6P, and UDP-GlcNAc (PDB 6SVP, [https://doi.org/10.2210/pdb6SVP/pdb]).

**In vitro PKA phosphorylation for Protein Mass Spectrometry**. PKA catalytic subunit from bovine heart (PKA, EC 2.7.11.11, Sigma-Aldrich P2645) was reconstituted in bi-distilled water containing 6 mg/ml DTT at a concentration of 50 μg/ml. Purified GFAT-1 variants were phosphorylated in an assay mixture containing 10 mM MgCl$_2$, 2 mM Na-ATP, and 20 U PKA in 100 μl GFAT-1 SEC buffer for 30 min at 30 °C. The samples were stored frozen. For proteomic analysis, 5 μg GFAT-1 was alkylated by 5 mM chloroacetamide, reduced with 1 mM TCEP, and digested by 0.1 μg Lys-C endoproteinase (MS Grade, Thermo Fisher Scientific) in 50 mM Tris/HCl, pH 8.3 overnight at 37 °C. The digest was acidified by addition of formic acid (end concentration 0.1%) and the resulting peptides were purified using C18 STAGE tips[70].

**Untargeted protein Mass Spectrometry**. 20% of STAGE tip purified peptides were separated on a 25 cm, 75 μm internal diameter PicoFrit analytical column (New Objective) packed with 1.9 μm ReproSil-Pur 120 C18-AQ media (Dr. Maisch HPLC GmbH) using an EASY-nLC 1200 (Thermo Fisher Scientific). The column was maintained at 50 °C. Buffer A and B were 0.1% formic acid in water and 0.1% formic acid in 80% acetonitrile. Peptides were separated on a segmented gradient from 6% to 31% buffer B for 57 min and from 31 to 44% buffer B for 5 min at 200 nl/min. Eluting peptides were analyzed on an Orbitrap Fusion Tribrid mass spectrometer (Thermo Fisher Scientific). Peptide precursor m/z was measured at 60,000 resolution in the 350 to 1500 m/z range. Precursors with charge state from 2 to 7 only were selected for HCD fragmentation using 27% normalized collision energy. The m/z values of the peptide fragments were measured at a resolution of 30,000 using an AGC target of 2e5 and 80 ms maximum injection time. Upon fragmentation, precursors were put on a dynamic exclusion list for 45 s.

Heavy synthetic peptides, corresponding to phosphorylated Ser205 were separated on a segmented gradient from 6 to 60% buffer B for 62 min at 200 nl/min. Analysis was carried out as described above.

Raw data were analyzed with MaxQuant version 1.6.1.0 using the integrated Andromeda search engine[71,72]. Peptide fragmentation spectra were searched against manually created GFAT-1 fasta file. The database was automatically complemented with sequences of contaminating proteins by MaxQuant. Methionine oxidation, protein N-terminal acetylation, and Phospho (STY) were set as variable modifications; cysteine carbamidomethylation was set as fixed modification. The digestion parameters were set to "specific" and "LysC/P," The minimum number of peptides and razor peptides for protein identification was 1; the minimum number of unique peptides was 0. Protein identification was performed at a peptide spectrum matches and protein false discovery rate of 0.01. For the analysis of the raw data from the synthetic heave peptides, Lys8 was set as "Standard" and used as a variable modification.

**Targeted protein Mass Spectrometry**. The phosphorylated (Ser205, pS), heavily labeled (K*) reference peptides Maxi SpikeTides L (wild type: SVHFPGQAVGTRRG-pS-PLLIGVRSEH-K* and mutant R203H:

SVHFPGQAVGTRHG-pS-PLLIGVRSEH-K*) were purchased from JPT Peptide Technologies (Berlin, Germany). Heavy peptides were dissolved in 50% acetonitrile, 0.05% formic acid in water and sonicated in a water bath for 1 min. The solution was further diluted using 0.1% formic acid in water for a final peptide concentration of 7 nM. For analysis, desalted GFAT-1 peptides were dissolved in 10 μl 0.1% formic acid in water and 2 μl of the peptide solution were combined with 2 μl of the tenfold diluted heavy peptide solution; 2 μl were analyzed by targeted mass spectrometry.

For targeted analysis, peptides were separated as described above. Targeted analysis was performed on an Orbitrap Fusion Tribrid mass spectrometer (Thermo Fisher Scientific). The m/z values for charge states 4 and 5 from the heavy and light peptide were chosen for targeted fragmentation across the entire LC run. The isolation width was 1.6 and collision energy was set to 27%. Fragment m/z values were measured in the Orbitrap in profile mode, at a resolution of 60 K, an AGC target of 5e4, and a maximum injection time of 118 ms.

Raw data was analyzed using Skyline[73] version 19.1.0.193. A library was built from the untargeted analysis of the heavy peptides. The library ion match tolerance was set to 0.05 m/z, the ten most intense product ions were picked. Ions with charges 1 or 2 and type y an b were used for quantification. An isotope modification of type "Label" was created using the following modification: 13C(6) 25N(2) (K). Isotope label type and internal standard type were set to "heavy". For quantification, MS level was set to 2.

**Thermal shift assay**. Protein thermal stability was analyzed by thermal shift (thermofluor) assays. Proteins were incubated with SYPRO orange dye (Sigma-Aldrich), which binds specifically to hydrophobic amino acids, elevating fluorescence at 610 nm when excited with a wavelength of 490 nm. The melting temperature is defined as the midpoint of temperature of the protein-unfolding transition[74]. This turning point of the melting curve was extracted from the derivative values of the RFU curve, where a turning point to the right is a minimum. Reactions were done in white RT-PCR plates and contained 5 μl SYPRO orange dye (1:500 dilution in ddH2O) and 5–10 μg protein in SEC buffer in a total volume of 50 μl. The melting curves were measured in a Bio-Rad CFX-96 Real-Time PCR machine at 1 °C/min at the FRET channel and the data analyzed with CFX Maestro (Bio-Rad).

**Circular dichroism (CD) spectroscopy**. GFAT-1 was dialyzed in 10 mM potassium phosphate buffer pH 7.5, 0.5 mM TCEP, 0.5 mM Na2Frc6P, and 10% (v/v) glycerol, and the protein concentration was adjusted to 0.2 mg/ml. UV spectra in the range of 195–260 nm were recorded with a J-715 CD spectropolarimeter (Jasco, Gross-Umstadt, Germany) at 20 °C using a quartz cuvette with 0.1 cm path length. The buffer baseline was recorded separately and subtracted from each sample spectrum. The obtained ellipticity (θ, deg) was converted to mean residue ellipticity [θ] using: $[\theta] = \theta/(10\ n\ c\ l)$ in deg cm$^2$ dmol$^{-1}$ ($n$ is the number of amino acids, $c$ the protein concentration in mol/l, and $l$ the path length of the cuvette).

**Fluorescence measurements**. Fluorescence emission spectra were measured with a FluoroMax®-4 (HORIBA) spectral fluorimeter at a concentration of 1.5 mg/ml in SEC buffer without Frc6P at 20 °C. Excitation wavelength was set to 295 nm and the spectra were recorded in a range of 305 to 450 nm in a quartz cuvette with 1.5 mm path length. The bandwidth for the excitation monochromator was 4 nm and for the emission monochromator 2 nm.

**Mammalian cell culture and stable cell line generation**. HEK293 cells (ATCC) were cultured at 37 °C, 5% CO2 on treated polystyrene culture dishes (Corning) in DMEM media with high glucose (4.5 g/l; 25 mM) with pyruvate (Gibco, 11995-065) supplemented with 10% fetal bovine serum (Gibco), 100 U/ml penicillin and 100 μg/ml streptomycin (Gibco).

Cells stably overexpressing GFAT-1 variants were generated by transfection of HEK293 cells with pcDNA3.1 plasmids with human GFAT-1 containing an internal His6-tag between Gly299 and Asp300 (pcDNA3.1-hGFAT1-His299). Internally tagged GFAT-1 was subcloned from pFL-hGFAT1-His299 using NheI and HindIII restriction sites. For each variant, one well of a 6 well plate was transfected with 2 μg of plasmid DNA with Lipofectamine® 2000 (LifeTechnologies™) according to the manufacturer's protocol. Selection was performed with 500 μg/ml G418 for several weeks. For Western blot analysis, cells were washed, and proteins were extracted with RIPA-buffer (120 mM NaCl, 50 mM Tris/HCl, 1% (v/v) NP40, 0.5% (w/v) deoxycholate, 0.1% (w/v) SDS; pH 7.5). Cell debris was removed by centrifugation, protein concentration of the supernatants were determined using Pierce™ bicinchoninic acid (BCA) protein assay kit (Thermo Scientific) according to the manufacturer's protocol. Two micrograms of cell lysate was separated by reducing SDS-PAGE and transferred to PVDF membranes. Primary antibodies against human GFAT1 and α-TUBULIN were used. Chemiluminescence of the appropriate secondary rabbit or mouse HRP-conjugated antibodies after incubation with ECL HRP substrate (Immobilon™ Western HRP Substrate, Millipore) was detected using a ChemiDoc™ Quantity One® system (Bio-Rad).

**Antibodies**. The following antibodies were used: GFAT1 (rb, EPR4854, Abcam ab125069, 1:1000), α-TUBULIN (ms, DM1A, Sigma T6199, 1:50,000), rabbit IgG (gt, LifeTechnologies G21234, 1:5000), and mouse IgG (gt, LifeTechnologies G21040, 1:5000).

**Reporting Summary**. Further information on research design is available in the Nature Research Reporting Summary linked to this article.

## Data availability

Structural data were deposited in the Protein Data Bank with the accession codes 6ZMJ [https://doi.org/10.2210/pdb6ZMJ/pdb] (GFAT-1 R203H), 6ZMK [https://doi.org/10.2210/pdb6ZMK/pdb] (GFAT-1 L405R), and 7NDL [https://doi.org/10.2210/pdb7NDL/pdb] (GFAT-1 S205D). The mass spectrometry proteomics data were deposited to the ProteomeXchange Consortium via the PRIDE[75] partner repository with the dataset identifier PXD020451. All other data supporting the presented findings are available from the corresponding authors upon request. Source data are provided with this paper.

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

## Acknowledgements

We thank all M.S.D. and U.B. laboratory members for helpful discussions. The AA2559 *gfat-1(dh783) C. elegans* strain and the Ahringer and Vidal RNAi libraries were kindly provided by Adam Antebi (Max Planck Institute for Biology of Ageing). The *syb3246 C. elegans* allele was generated by SunyBiotech (China). Whole genome sequencing was done at the Cologne Center for Genomics. We thank Yvonne Hinze and Patrick Giavalisco from the metabolomics core facility of the Max Planck Institute for Biology of Ageing. We are grateful to Schirin Birkmann for support in the insect cell maintenance. We thank Stephan Miethe for help with the mammalian cell culture experiments. Crystals were grown in the Cologne Crystallization facility (C₂f). We thank the staff of beamline X06DA at the Swiss Light Source, Paul Scherrer Institute, Villigen (Switzerland) and beamline P13 at PETRA III, DESY, Hamburg (Germany) for their support during

data collection. This work was supported by the German Federal Ministry of Education and Research (BMBF, grant 01GQ1423A EndoProtect), by the German Research Foundation (DFG, Projektnummer 73111208-SFB 829, B11 and B14), by the European Commission (ERC-2014-StG-640254-MetAGEn), and by the Max Planck Society. The Cologne Crystallization Facility C$_2$f was supported by DFG grant INST 216/949-1 FUGG.

## Author contributions

S.R. and M.S.D. designed the project. S.R. performed the biochemical and crystallization experiments, as well as the cell culture experiments. F.A.M.C.M. performed the experiments related to *C. elegans*. I.A. did the protein mass spectrometry measurements and analysis. S.R., M.S.D., and U.B. wrote the manuscript. S.R. prepared the figures.

## Funding

## Competing interests

The authors declare no competing interests.
