## [Peer Review File · Nature Communications]

REVIEWER COMMENTS

Reviewer #1 (Remarks to the Author):

The manuscript NCOMMS-20-30151, titled „Protein kinase A controls the hexosamine pathway by tuning the feedback inhibition of GFAT1“, is an heroic effort by the authors to shed light on the cause of the gain of function by a single site mutation R203H in GFAT-1.

The manuscript is very interesting and of particular importance to understand how small changes introduced by PTM's have profound consequences on protein function. However, often it feels that arguments are slightly contradictory and it does not feel compelling and very logical in its current form. A lot of the experiments are done on the phosphomimic, but these results seem not to align very well with the trend on the phosphorylated protein *in vivo*. In its current form, I cannot recommend the publication unreservedly. Though I believe most of my criticism in respect of the results and conclusions derived can be addressed without additional experiments. Some of my questions are written below in more detail, which hopefully show my concerns.

Introduction

Line 44

Relative orientation

Line 53

The initial anchoring is not a pure rotation I presume? Or maybe that is not clear. The second subsequent rotation of 21° seems very precise. Anyway a slight rewrite of that two sentences would be good to get a better understanding that first the domain is locked and upon substrate binding rotates again by 21°.

Line 69 No citations for phosphorylation?

Line 80-82 First it is said the function of Ser243 phosphorylation is elusive but then it is stated that it might be activating or inhibiting. So there is an effect and it was measured, or not?

Line 85 to 100: That is not written very clear. I presume that the mutations result in a gain of function due to the lack of feedback inhibition. I suppose that the authors infer from activity assays that both domains are affected, but UDPGlcNAc binds just to one domain. Now we come to a second gain of function mutation, which influences phosphorylation I believe.?

That is very confusing in its current form.

Results

Line 122 onwards

From a logical point of view, wouldn't one want to characterize the mutation first and then crystallise the protein. I know the chronological flow of experiments is often different, but I would first give the confirmation that in humans the mutation does something similar, before diving into the crystallographic results, to get a better understanding.

I cannot see from Fig2b that the loop 277-80 is stabilized. At least D279 seems pretty much in the exact same orientation.

Line 164. It was mentioned that there are conflicting results regarding the effect of phosphorylation. However, it is clearly argued that reduced phosphorylation is a gain of function mechanism. On which particular result is that claim based? Some of this is revealed in the next paragraph

Line 187 onwards

It is shown that the catalytic activity is influenced for both domains. Furthermore, the S205D variant

has no feedback inhibition at all. How does phosphorylation at low concentration of UDP-GlcNAc inhibit the enzyme? Phosphorylation reduces kinetic constant and abolishes feedback inhibition completely. Therefore, the enzyme is just constantly active, or does that refer to a change in K_m ?

Line 199 It seems from the results in HEK cells that phosphorylation does not play a significant role, despite the functional S205D variant. Does it mean that there is no sufficient phosphorylation *in vivo*? Also there is no significant change in GlcN/GalN levels, but there is a measurable change in the worm. Does that not point to differences in regulation between the human and the *C. elegans* GFAT-1?

Discussion:

Line 273: I am confused. The S205 D mutant shows there is no feedback inhibition, but lower activity (kinetic constants). That means with phosphorylation, it is expected that the enzyme has no feedback inhibition. How does the reduced phosphorylation due to the mutation decrease the feedback inhibition?

In general, all conclusions are based on the S205D mimic, However at least some assays could have been done with phosphorylated GFAT as there was an *in vitro* phosphorylation assay. For example, the melting point assays could have been done with the PKA treated protein.

Especially the experiments with overexpression of GFAT-1 and the mutants didn't show any effect on GlcN levels, except for the mimic, which might be as well an effect of a more specific/stable saltbridge similar to L405R.

The different behaviour at low and high levels of GlcNAc is not clear to me, or it is somehow written in a very convoluted way.

Reviewer #2 (Remarks to the Author):

The paper reports the regulation of GFAT-1 by PKA at critical sites in the GFAT coding region. Of note, Ser205 phosphorylation had two discernible effects: It lowered baseline GFAT-1 activity and abolished UDP-GlcNAc feedback inhibition. Thus, PKA

controls the HP by uncoupling the metabolic feedback loop of GFAT-1. They describe that the R203H mutation influences 205 phosphorylation which is the target of PKA. However, their biochemical data (kinetic data) is not similar between R203H and S205D as summarized in table 2 (Fig. 2c,d,e and fig. 4a,b, c,e).

The level of PKA phosphorylation defect in R203H mutant is ~60% reduction compared to WT. But the concentration of UDP-GlcNAc does not change between WT and R203H in HEK293 cell. This observation is not quite the same in the *C. elegans* *gfat-1(dh783)* mutant, which decreases the level of UDP-GlcNAc concentration (over twice). In their original work, It seems that they originally cloned about three alleles of *gfat-1(gof)* mutant. They describe the fourth allele of *gfat-1(gof)* in current paper. They may need to confirm that this R203H mutation indeed causes gain of function of *gfat-1*, by created the same mutation by using Crispr/cas9. However, it not convincing that *C. elegans* (R203H) mutation has similar function in Human *gfat1*. They may need to use *C. elegans* *gfat-1* (WT, R203H and S205D) for their biochemical assay.

The authors argue that:

A new GFAT-1 gain-of-function point mutant is resistant to ER stress

The GFAT-1 R203H substitution interferes with UDP-GlcNAc inhibition leading to gain-of-function

The GFAT-1 R203H substitution interferes with PKA phosphorylation at Ser205

PKA phosphorylation at Ser205 modulates UDP-GlcNAc inhibition of GFAT-1

PKA phosphorylation at Ser205 modulates GFAT-1 domain stability

The authors suggest that the GFAT-1 mutant R203H and decipher its two gain-of-function mechanisms: (1) strongly reduced PKA-dependent phosphorylation at Ser205 due to a disrupted consensus sequence and (2) a weaker sensitivity to UDP-GlcNAc feedback inhibition

These observations are interesting given the constraints of the experiments suggested above

Reviewer #3 (Remarks to the Author):

This manuscript discusses the mechanism by which the activity of the GFAT enzyme is regulated by serine phosphorylation and UDP-GlcNAc feedback inhibition. In particular, the authors showed that phosphorylation of S205 decreases the basal activity of GFAT and abolishes UDP-GlcNAc feedback inhibition. Therefore, at low concentration of UDP-GlcNAc, phosphorylation is inhibitory, while, at high concentration of UDP-GlcNAc, phosphorylation stimulates GFAT activity. The authors move on to investigate the structural basis of such observation. They do so by studying the effect of phosphorylation (or better of the S205D phosphomimetic mutant) on the thermal stability of the enzyme. They show that the S205D mutation lowers the melting temperature (T_m) of the glutaminase domain and leaves the T_m of the isomerase domain unperturbed. From these data, the authors infer that phosphorylation affects the relative positioning of the domains by altering their catalytic ability and their functional response to UDP-GlcNAc binding.

I have mixed feelings about this manuscript. I really enjoyed the functional characterization of the enzyme and of its elegant regulatory mechanism. However, I think the proposed structural basis for precisely how S205 phosphorylation affects GFAT activity is not properly supported by the experimental data. Indeed, as S205 is located in the glutaminase domain, it is very likely that the S205D mutant lowers the intrinsic stability of the glutaminase domain, without having to invoke an interdomain rearrangement. Indeed, while perturbing the domain-domain interaction by adding salts or by introducing an interdomain salt-bridge (via the L405R mutation) affects the T_m of both domains, the S205D mutant only destabilizes the glutaminase domain, leaving the T_m of the isomerase domain unperturbed.

It is the reviewer opinion that further data need to be acquired to proof the proposed model. In this respect FRET or EPR data could be very informative, but any other experiment able to probe the relative orientation between the domains would be sufficient

RESPONSES TO REVIEWER COMMENTS

Reviewer #1 (Remarks to the Author):

The manuscript NCOMMS-20-30151, titled „Protein kinase A controls the hexosamine pathway by tuning the feedback inhibition of GFAT1”, is an heroic effort by the authors to shed light on the cause of the gain of function by a single site mutation R203H in GFAT-1.

The manuscript is very interesting and of particular importance to understand how small changes introduced by PTM's have profound consequences on protein function. However, often it feels that arguments are slightly contradictory and it does not feel compelling and very logical in its current form. A lot of the experiments are done on the phosphomimic, but these results seem not to align very well with the trend on the phosphorylated protein in vivo. In its current form, I cannot recommend the publication unreservedly. Though I believe most of my criticism in respect of the results and conclusions derived can be addressed without additional experiments.

Some of my questions are written below in more detail, which hopefully show my concerns.

We thank the reviewer for the positive assessment! Further, we are grateful for the valuable comments that led us to rewrite essential passages of the manuscript. The in vitro and in vivo data of the S205D mutation are consistent when considering that in cells the UDP-GlcNAc concentration is high enough to inhibit WT GFAT-1. We have taken great care to better bring out these observations.

Introduction

Line 44

Relative orientation

Thanks for the comment, we have specified the orientation.

Line 53

The initial anchoring is not a pure rotation I presume? Or maybe that is not clear. The second subsequent rotation of 21° seems very precise. Anyway a slight rewrite of that two sentences would be good to get a better understanding that first the domain is locked and upon substrate binding rotates again by 21°.

We thank the reviewer for pointing this out and have rephrased our manuscript. Without substrate binding, the glutaminase domain is very flexible and no corresponding electron density is visible in the E. coli apo structure. Given that the structural state of the glutaminase domain before Frc6P binding is not known, the changes upon Frc6P binding cannot exactly be determined.

Line 69

No citations for phosphorylation?

We have edited the manuscript accordingly.

Line 80-82

First it is said the function of Ser243 phosphorylation is elusive but then it is stated that it might be activating or inhibiting. So there is an effect and it was measured, or not?

The literature is contradictory regarding the effect of PTMs of GFAT-1 on its activity. We stated that it is elusive as there are contradicting observations. We have edited these sentences to be clearer: "Ser243 was shown to be phosphorylated by adenosine monophosphate (AMP)-activated protein kinase (AMPK) and calcium/calmodulin-dependent kinase II, but the effect on GFAT activity is not clear: depending on the study, phosphorylation at this site is reported to be either activating or inhibiting."

Line 85 to 100:

That is not written very clear. I presume that the mutations result in a gain of function due to the lack of feedback inhibition. I suppose that the authors infer from activity assays that both domains are affected, but UDPGlcNAc binds just to one domain. Now we coming to a second gain of function mutation, which influences phosphorylation I believe.? That is very confusing in its current form.

We have edited this passage to be clearer (lines 86f).

Results

Line 122 onwards

From a logical point of view, wouldn't one want characterize the mutation first and then crystallise the protein. I know the chronological flow of experiments is often different, but I would first give the confirmation that in humans the mutation does something similar, before diving into the crystallographic results, to get a better understanding.

We implemented these changes and have rearranged the order of this figure.

I cannot see from Fig2b that the loop 277-80 is stabilized. At least D279 seems pretty much in the exact same orientation.

We agree that our statement was misleading and thank the reviewer for pointing this out. We changed the sentence to: "the side chain of His203 protrudes between Asp278 and Asp279, without affecting the orientation of the neighboring loop (residues 277-280)".

Line 164. It was mentioned that there are conflicting results regarding the effect of phosphorylation. However, it is clearly argued that reduced phosphorylation is a gain of function mechanism. On which particular result is that claim based? Some of this is revealed in the next paragraph

We thank the reviewer for pointing this out. We deleted the conclusion for this section that loss of Ser205 phosphorylation is a gain-of-function mechanism and have re-written this passage to highlight that the R203H substitution cause a loss of PKA-dependent regulation.

Line 187 onwards

It is shown that the catalytic activity is influenced for both domains. Furthermore, the S205D variant has no feedback inhibition at all. How does phosphorylation at low concentration of UDP-GlcNAc inhibit the enzyme? Phosphorylation reduces kinetic constant and abolishes feedback inhibition completely. Therefore, the enzyme is just constantly active, or does that refer to a change in Km?

At this point we have to distinguish between the interpretation of the consequences of phosphorylation *in vitro* and *in vivo*. We are grateful to the reviewer for pointing out the weakness in our explanation and re-phrased our interpretation to make this clearer (lines 196f). *In vitro*, the phosphorylation lowers the catalytic activity and abolishes the UDP-GlcNAc feedback inhibition. *In vivo*, UDP-GlcNAc is present and the concentration of UDP-GlcNAc determines whether the lower activity or the loss of UDP-GlcNAc inhibition predominantly affect the activity of GFAT-1. At low UDP-GlcNAc concentrations, the GlcN6P synthesis is lower in the Ser205 phosphorylated GFAT-1 than in unphosphorylated wild type GFAT-1 due to reduced catalytic activity. At high UDP-GlcNAc concentrations (on the right side of the curve, below), phospho-Ser205 GFAT-1 is constitutively active due to loss of inhibition and therefore more active than unphosphorylated GFAT-1.

Line 199

It seems from the results in HEK cells that phosphorylation does not play a significant role, despite the functional S205D variant. Does it mean that there is no sufficient phosphorylation *in vivo*?

In our in vitro phosphorylation assays we could reproducibly phosphorylate only <20% of the purified GFAT-1 (please also see our comment regarding assays with in vitro phosphorylated GFAT-1 below). Given that Ser205 is in the interface between both domains, this site might not be well accessible and therefore not completely phosphorylated in vivo. We also noted that we could only achieve a mild overexpression of the S205D variant in HEK cells compared to the strong overexpression of WT or the R203H GFAT-1 (Fig. 4d). Moreover, we could only obtain a heterozygous S202D worm (C. elegans S202D is homologous to human S205D) and not a homozygous C. elegans strain (lines 219f). Together, this suggests a detrimental effect of a very strong GFAT-1 activation through loss of feedback inhibition. While a mild stimulation of the hexosamine pathway through PKA phosphorylation at Ser205 might be beneficial, a complete lack of UDP-GlcNAc dependent regulation might be not desirable. In line with this, the strong overexpression of WT GFAT-1 does not lead to increased UDP-GlcNAc levels. This is important as it shows that WT GFAT-1 is under very powerful inhibition by UDP-GlcNAc. While we assume that kinases are present and active in the HEK cells, under replete conditions, there is no strong regulation of GFAT-1.

Also there is no significant change in GlcN/Galn levels, but there is a measurable change in the worm. Does that not point to differences in regulation between the human and the C. elegans GFAT-1?

We agree that there is no measurable change in HEK cells when WT GFAT-1 is overexpressed. In the worm, however, we did not perform a comparable experiment. The HEK cells were cultured at high glucose concentrations (25 mM Glc), while the physiological glucose concentrations in the worm are, conceivably, lower. Moreover, the respective worm strains all carry point mutations in the endogenous GFAT-1 locus. In fact, all our data point to a very strong degree of conservation of GFAT-1 and its regulation. To further support this, we

generated the S202D mutation in the worm that corresponds to the human Ser205 position. This mutant showed significant tunicamycin resistance, consistent with elevated UDP-GlcNAc levels (Fig. 4f).

Discussion:

Line 273: I am confused. The S205D mutant shows there is no feedback inhibition, but lower activity (kinetic constants). That means with phosphorylation, it is expected that the enzyme has no feedback inhibition. How does the reduced phosphorylation due to the mutation decrease the feedback inhibition?

The reviewer is correct, the reduced phosphorylation in the R203H mutant did not decrease the feedback inhibition. We rephrased this sentence to make this clearer. We have to distinguish between two independent gain-of-function mechanisms, which are influenced by the UDP-GlcNAc concentrations and are explained in line 322 onwards.

In general, all conclusions are based on the S205D mimic, However at least some assays could have been done with phosphorylated GFAT as there was an in vitro phosphorylation assay. For example, the melting point assays could have been done with the PKA treated protein.

We thank the reviewer for this nice suggestion and have ourselves aimed at using an in vitro phosphorylated GFAT-1 for experiments. In vitro phosphorylation, however, turned out to have significant drawbacks. First, several other sites were phosphorylated upon PKA treatment (Supplementary Dataset 1). Therefore, effects seen with in vitro phosphorylated GFAT-1 could not clearly be assigned to Ser205 phosphorylation. We generated a GFAT-1 S235A to control at least for the second known PKA site. However, second, when we performed in vitro PKA phosphorylation of wild type or S235A GFAT-1, only a fraction (< 20%) of GFAT-1 was found phosphorylated:

Ser205 phosphorylation occupancy in wild type or S235A GFAT-1 upon PKA-treatment for 30 to 120 min. The amount of phosphorylation was measured by MS with a heavy isotope labeled phosphorylated reference peptide. Two different charges of the peptide were identified in MS.

Based on these complications, we are showing the data using the S205D substitution that is the most specific means to investigate the consequences of phosphorylation at that site.

Especially the experiments with overexpression of GFAT-1 and the mutants didn't show any effect on GlcN levels, except for the mimic, which might be as well an effect of a more specific/stable saltbridge similar to L405R.

We thank the reviewer for pointing this out. In fact, the observation that GFAT-1 overexpression did not lead to elevated UDP-GlcNAc levels is a salient point here. WT GFAT-1 is effectively inhibited. Ser205 modification, however, likely prevents this inhibitory function of UDP-GlcNAc. We further agree that the salt bridges are the likely mechanism by which domain interaction, inhibition by UDP-GlcNAc, and the consequence of Arg203 and Ser205 are linked.

The different behaviour at low and high levels of GlcNAc is not clear to me, or it is somehow written in a very convoluted way.

We agree that the different behavior had to be clarified and we explained it in lines 196f. The data refer to Fig. 4c in which we observe two effects of the S205D substitution. At low UDP-GlcNAc levels, there is a lower baseline enzymatic activity, while at high UDP-GlcNAc levels (that inhibit WT activity), the S205D variant remains unaffected.

Reviewer #2 (Remarks to the Author):

The paper reports the regulation of GFAT-1 by PKA at critical sites in the GFAT coding region. Of note, Ser205 phosphorylation had two discernible effects: It lowered baseline GFAT-1 activity and abolished UDP-GlcNAc feedback inhibition. Thus, PKA controls the HP by uncoupling the metabolic feedback loop of GFAT-1. They describe that the R203H mutation influences 205 phosphorylation which is the target of PKA. However, their biochemical data (kinetic data) is not similar between R203H and S205D as summarized in table 2 (Fig. 2c,d,e and fig. 4a,b, c,e).

The level of PKA phosphorylation defect in R203H mutant is ~60% reduction compared to WT. But the concentration of UDP-GlcNAc does not change between WT and R203H in HEK293 cell. This observation is not quite the same in the *C.elegans* *gfat-1*(dh783) mutant, which decreases the level of UDP-GlcNAc concentration (over twice). In their original work, It seems that they originally cloned about three alleles of *gfat-1*(gof) mutant. They describe the fourth allele of *gfat-1*(gof) in current paper. They may need to confirm that this R203H mutation indeed causes gain of function of *gfat-1*, by created the same mutation by using Crispr/cas9. However, it not convincing that *C. elegans* (R203H) mutation has similar function in Human *gfat1*. They may need to use *C. elegans* *gfat-1*(WT, R203H and S205D) for their biochemical assay.

We appreciate these valuable comments and have carefully addressed them. The PKA phosphorylation defect in the R203H variant reduces phosphorylation by a factor of 80 (line 169). We have juxtaposed the two bar graphs with different scales of the Y-axis to highlight this difference. We adjusted the figure to make this clearer (Fig. 3d). This shows that R203H interferes with the phosphorylation at Ser205.

*This is consistent with the data from the HEK cells, where un-phosphorylated GFAT-1 can be fully inhibited by UDP-GlcNAc. The *gfat-1*(dh783) mutant shows significant elevation of UDP-GlcNAc, consistent with a gain-of-function and weaker UDP-GlcNAc mediated inhibition (Fig. 4e).*

*The reviewer is correct in the statement that here we are showing a fourth GFAT-1 gain-of-function mutant. We have now performed a number of experiments that further support the conclusion that the dh783 allele is in fact acting via GFAT-1 gain-of-function. First, we asked if the hexosamine pathway is responsible for the observed effects. We knocked down GNA-2 that is responsible for the acetylation step of the hexosamine pathway and found that the RNAi treatment significantly reduced the dh783 tunicamycin resistance (Fig. 1e). Second, the UDP-GlcNAc and UDP-GalNAc levels are elevated in dh783 compared to wild type (Fig. 1f, Supplementary Fig. 1b). Thus, we confirmed that the R203H mutation indeed caused the *gfat-1* gain-of-function.*

*To address the reviewer's last point, we have recombinantly expressed *C. elegans* GFAT-1, but unfortunately, most protein was insoluble, yielding a low protein amount after a two-step purification. Moreover, the purified *C. elegans* GFAT-1 preparation included two species. Therefore, we were unable to use nematode GFAT-1 for in vitro assays.*

To control for the reviewer's concerns regarding the conservation between *C. elegans* and human GFAT-1, we decided to introduce the S205D mutation into the worm GFAT-1 and generated the (homologous) *C. elegans* GFAT-1 S202D, using CRISPR/Cas9. Consistent with a failure of inhibition by UDP-GlcNAc, as observed in the human S205D variant, these worms were tunicamycin resistant (Fig. 4f).

Together these new data support the notion that *dh783* is in fact an allele of GFAT-1, leading to gain-of-function and that the consequences of phosphorylation are conserved.

The authors argue that:

A new GFAT-1 gain-of-function point mutant is resistant to ER stress

The GFAT-1 R203H substitution interferes with UDP-GlcNAc inhibition leading to gain-of-function

The GFAT-1 R203H substitution interferes with PKA phosphorylation at Ser205

PKA phosphorylation at Ser205 modulates UDP-GlcNAc inhibition of GFAT-1

PKA phosphorylation at Ser205 modulates GFAT-1 domain stability

The authors suggest that the GFAT-1 mutant R203H and decipher its two gain-of-function mechanisms: (1) strongly reduced PKA-dependent phosphorylation at Ser205 due to a disrupted consensus sequence and (2) a weaker sensitivity to UDP-GlcNAc feedback inhibition

These observations are interesting given the constraints of the experiments suggested above

We thank the reviewer for pointing out the key findings and conclusions of our manuscript. As detailed above, we have addressed the reviewers concerns with new data sets that strengthen our original conclusions.

Reviewer #3 (Remarks to the Author):

This manuscript discusses the mechanism by which the activity of the GFAT enzyme is regulated by serine phosphorylation and UDP-GlcNAc feedback inhibition. In particular, the authors showed that phosphorylation of S205 decreases the basal activity of GFAT and abolishes UDP-GlcNAc feedback inhibition. Therefore, at low concentration of UDP-GlcNAc, phosphorylation is inhibitory, while, at high concentration of UDP-GlcNAc, phosphorylation stimulates GFAT activity. The authors move on to investigate the structural basis of such observation. They do so by studying the effect of phosphorylation (or better of the S205D phosphomimetic mutant) on the thermal stability of the enzyme. They show that the S205D mutation lowers the melting temperature (T_m) of the glutaminase domain and leaves the T_m of the isomerase domain unperturbed. From these data, the authors infer that phosphorylation affects the relative positioning of the domains by altering their catalytic ability and their functional response to UDP-GlcNAc binding.

I have mixed feelings about this manuscript. I really enjoyed the functional characterization of the enzyme and of its elegant regulatory mechanism. However, I think the proposed structural basis for precisely how S205 phosphorylation affects GFAT activity is not properly supported by the experimental data. Indeed, as S205 is located in the glutaminase domain, it is very likely that the S205D mutant lowers the intrinsic stability of the glutaminase domain, without having to invoke an interdomain rearrangement. Indeed, while perturbing the domain-domain interaction by adding salts or by introducing an interdomain salt-bridge (via the L405R mutation) affects the T_m of both domains, the S205D mutant only destabilizes the glutaminase domain, leaving the T_m of the isomerase domain unperturbed.

It is the reviewer opinion that further data need to be acquired to proof the proposed model. In this respect FRET or EPR data could be very informative, but any other experiment able to probe the relative orientation between the domains would be sufficient

We thank the reviewer for the valuable feedback for our manuscript and appreciate the comments aimed to improve the paper. Prompted by the reviewer's comment we initiated two independent experiments to further analyze the structural consequence of the S205D substitution and to test our proposed model. First, we generated a new crystal structure of GFAT-1 S205D and second, we analyzed the domain interactions by tryptophan fluorescence.

GFAT-1 S205D did not crystallize in the standard human GFAT-1 crystallization condition, but we identified and optimized a new crystallization condition for GFAT-1 S205D. We solved the structure of S205D and included it in Fig. 5a. Within the crystal structure, we did not observe a changed positioning of the glutaminase domain relative to the isomerase domain, when compared to wild type GFAT-1. These data are not necessarily contradictory to our model of changed domain interactions and possible domain movements, because we captured only one structural state within the crystal. Moreover, the crystallographic B-factors are very similar between the wild type and S205D structure (Supplementary Fig. 5m,n), which argues against the reviewer's notion that the reduced intrinsic stability of the glutaminase domain most readily explains the effects on activity and feedback.

Additionally, fluorescence emission spectra were measured in solution. In GFAT-1, Trp96 is between the two domains at the glutaminase active site and seems to be optimally positioned to monitor domain movement (Fig. 5e). Given that GFAT-1 possess a second tryptophan, Trp50, that might mask an effect of Trp96, we mutated Trp50 to phenylalanine (W50F). Strikingly, we found significant changes in the fluorescence spectra of W50F and the double mutant W50F/S205D (Fig. 5f, g). These measurements indicate a changed environment of Trp96 when comparing wild type and S205D GFAT-1 and thus provide some support to a model of phosphorylation-dependent domain-movement.

Ultimately, available methods are not going to provide a definitive answer to the question if phospho-Ser205 leads to a slight re-arrangement of the two GFAT-1 domains. We have followed the reviewer's suggestion and discuss domain-domain interactions in more broad terms across the manuscript and use the discussion to speculate about possible changes in domain orientation as this was also suggested by our previously published observations.

REVIEWERS' COMMENTS

Reviewer #1 (Remarks to the Author):

The manuscript titled "Protein Kinase A controls the hexosamine pathway by tuning the feedback inhibition of GFAT-1" is a resubmission after revision. The authors present in the manuscript an even more impressive amount of data to resolve the molecular determinants of the R203H gain of function, as well as the role of the phosphorylation of S205D.

Most of the comments are addressed in the resubmission and there is just one comment/question, I would like to ask just out of interest. There is no need to address this in the manuscript as a must. The gain of function for R203H is due to a higher IC50 of for UDP-GlcNAc as inhibitor. The concentration seems high enough in the HEK293 experiment to inhibit both wt and mutant very efficient. The authors measured the GlcNAc/GalNAc concentration in the cells. However, it is related to GlcNAc per mg protein. Can one somehow roughly estimate how much that would be in μM in the cell? That should be at least higher than 250 μM , to ensure complete inhibition, I presume.

Everything else was addressed in the revision and from my side the manuscript is ready for publication

Reviewer #2 (Remarks to the Author):

My concerns were addressed

Reviewer #3 (Remarks to the Author):

The authors addressed my concerns

RESPONSES TO REVIEWER COMMENTS

Reviewer #1 (Remarks to the Author):

The manuscript titled "Protein Kinase A controls the hexosamine pathway by tuning the feedback inhibition of GFAT-1" is a resubmission after revision. The authors present in the manuscript an even more impressive amount of data to resolve the molecular determinants of the R203H gain of function, as well as the role of the phosphorylation of S205D.

Most of the comments are addressed in the resubmission and there is just one comment/question, I would like to ask just out of interest. There is no need to address this in the manuscript as a must.

The gain of function for R203H is due to a higher IC₅₀ of for UDP-GlcNAc as inhibitor. The concentration seems high enough in the HEK293 experiment to inhibit both wt and mutant very efficient. The authors measured the GlcNAc/GalNAc concentration in the cells. However, it is related to GlcNAc per mg protein. Can one somehow roughly estimate how much that would be in μM in the cell? That should be at least higher than 250 μM , to ensure complete inhibition, I presume.

Everything else was addressed in the revision and from my side the manuscript is ready for publication

We thank the reviewer for the supportive evaluation of our manuscript. Throughout the project, we have wondered about cytoplasmic UDP-GlcNAc concentrations and based on our observations the reviewer is pointing out, we are coming to a similar conclusion. Direct cytoplasmic concentration measurement would be highly relevant but is currently not possible. Certainly, our data support a model in which the R203H variant is fully inhibited in HEK293 cells.

Reviewer #2 (Remarks to the Author):

My concerns were addressed

Reviewer #3 (Remarks to the Author):

The authors addressed my concerns

We thank reviewers #2 and #3 for their input to our manuscript and their constructive feedback during the revision.